# Induction of the IL-1RII decoy receptor by NFAT/FOXP3 blocks IL-1β-dependent response of Th17 cells

Dong Hyun Kim[1], Hee Young Kim[1,2,3], Sunjung Cho[2], Su-Jin Yoo[4], Won-Ju Kim[5], Hye Ran Yeon[6], Kyungho Choi[6], Je-Min Choi[5], Seong Wook Kang[4], Won-Woo Lee[1,2,3,7]*

[1]Laboratory of Autoimmunity and Inflammation (LAI), Department of Biomedical Sciences, Seoul National University College of Medicine, Seoul, Republic of Korea; [2]Department of Microbiology and Immunology, Seoul National University College of Medicine, Seoul, Republic of Korea; [3]Cancer Research Institute and Institute of Infectious Diseases, Seoul National University College of Medicine, Seoul, Republic of Korea; [4]Department of Internal Medicine, Chungnam National University School of Medicine, 282 Munhwa-ro, Jung-gu, Daejeon, Republic of Korea; [5]Department of Life Science, College of Natural Sciences and Research Institute for Natural Sciences, Hanyang University, Seoul, Republic of Korea; [6]Department of Biochemistry and Molecular Biology, Department of Biomedical Sciences, and Cancer Research Institute, Seoul National University College of Medicine, Seoul, Republic of Korea; [7]Ischemic/Hypoxic Disease Institute, Seoul National University College of Medicine; Seoul National University Hospital Biomedical Research Institute, Seoul, Republic of Korea

**Abstract** Derived from a common precursor cell, the balance between Th17 and Treg cells must be maintained within immune system to prevent autoimmune diseases. IL-1β-mediated IL-1 receptor (IL-1R) signaling is essential for Th17-cell biology. Fine-tuning of IL-1R signaling is controlled by two receptors, IL-1RI and IL-RII, IL-1R accessory protein, and IL-1R antagonist. We demonstrate that the decoy receptor, IL-1RII, is important for regulating IL-17 responses in TCR-stimulated CD4[+] T cells expressing functional IL-1RI via limiting IL-1β responsiveness. IL-1RII expression is regulated by NFAT via its interaction with Foxp3. The NFAT/FOXP3 complex binds to the *IL-1RII* promoter and is critical for its transcription. Additionally, IL-1RII expression is dysregulated in CD4[+] T cells from patients with rheumatoid arthritis. Thus, differential expression of IL-1Rs on activated CD4[+] T cells defines unique immunological features and a novel molecular mechanism underlies IL-1RII expression. These findings shed light on the modulatory effects of IL-1RII on Th17 responses.

*For correspondence: wonwoolee@snu.ac.kr

Competing interests: The authors declare that no competing interests exist.

## Introduction

Interleukin-1 (IL-1) consists of two distinct forms (IL-1α and IL-1β) and is a master cytokine of local and systemic inflammation (*Dinarello et al., 2012*). Primarily produced by monocytes, macrophages, and dendritic cells (DCs) in response to stimulation, IL-1β is involved in a variety of biological activities such as cell proliferation, differentiation, and apoptosis through its binding to functional IL-1 receptor I (IL-1RI) expressed on virtually all cells. Thus, IL-1β plays a role as a critical pathogenic mediator of many autoinflammatory, autoimmune, infectious, and degenerative diseases (*Dinarello, 2011*).

In human T-cell immunity, IL-1β has been identified as an essential cytokine with a role similar to that played by IL-6 in mice for Th17 differentiation, and is also important for expansion, in vivo survival, and effector function of IL-17-producing T cells (*Acosta-Rodriguez et al., 2007*; *Lee et al., 2010*). More recently, the importance of IL-1 signaling has been also emphasized in murine Th17 cells. Even though TGF-β and IL-6 are sufficient for murine Th17 differentiation, the IL-1RI knockout (KO) mouse is resistant to experimental autoimmune encephalomyelitis due to a failure to generate Th17 cells (*Sutton et al., 2006*; *Chung et al., 2009*). Further, IL-1β and IL-23 along with IL-6 are critical for generation of highly pathogenic Th17 cells in an autoimmune mouse model (*Lee et al., 2012*; *Ronchi et al., 2016*). Recent studies have demonstrated that IL-1β functions as a proinflammatory regulator of pathogenic human Th17 cells, which concomitantly produce IFN-γ and GM-CSF (*Zielinski et al., 2012*). Due to its potent effect as a proinflammatory cytokine, the biological activity of IL-1β is tightly regulated at multiple levels by diverse mechanisms including receptor antagonism through production of a decoy receptor and soluble forms of signaling receptors or accessory proteins (*Garlanda et al., 2013a*).

The IL-1R signaling complex includes the functional IL-1RI subunit and a signaling subunit, interleukin-1 receptor accessory protein (IL-1RAcP). Dimerization of the IL-1R1 and IL-1RAcP cytoplasmic Toll/IL-1R (TIR) domains initiates a signaling cascade through recruitment of adapter proteins (*Garlanda et al., 2013b*). The decoy IL-1RII has a short cytoplasmic tail lacking the signal-transducing TIR domain and acts as a molecular trap, capturing IL-1 with high-affinity and exerting a dominant-negative effect (*Shirakawa et al., 1987*). In contrast to ubiquitously expressed IL-1RI, the expression of IL-1RII is restricted to cells including monocytes, neutrophils, and B cells (*Martin et al., 2017*; *Colotta et al., 1996*; *Colotta et al., 1993*; *McMahan et al., 1991*). In addition, it was recently demonstrated that human-activated Treg cells and follicular regulatory T (T$_{fr}$) cells upregulate IL-1RII to attenuate their IL-1-dependent responses (*Tran et al., 2009*; *Ritvo et al., 2017*), suggesting a physiological significance for IL-1RII expression in immune regulation.

The physiological relevance of IL-1RII expression was investigated in autoimmune arthritis animal models using an IL-1RII conditional KO mouse. Deficiency of IL-1RII was found to exacerbate autoimmune arthritis by inhibiting IL-1 signaling in innate cells, such as macrophages and neutrophils (*Martin et al., 2017*; *Shimizu et al., 2015*). Our previous study showed that IL-1RII mRNA expression is markedly higher in IL-1RI$^+$ memory CD4$^+$ T cells compared to IL-1RI$^-$ memory CD4$^+$ T cells. Further, the negative regulatory role of IL-1RII was further evidenced by the significant augmentation of IL-17 production by IL-1RI$^+$ memory CD4$^+$ T cells in response to IL-1β and TCR stimulation after treatment with neutralizing IL-1RII antibody (*Lee et al., 2010*). Given its preferential expression on IL-17-producing IL-1RI$^+$ CD4$^+$ T cells as well as Tregs, IL-1RII has been suggested to play a pivotal role in maintaining the balance between Th17 and Treg cells. These cells are reciprocally interconnected in their development pathways and conversion between the two subsets is critical for pathogenesis and resolution of many inflammatory disorders (*Zielinski et al., 2012*; *Basu et al., 2015*; *Gagliani et al., 2015*). However, little is known about the molecular mechanism underlying IL-1RII expression by CD4$^+$ T cells and its contribution to modulation of Th17 responses in humans.

Here, we provide novel insight into the immune regulatory role of decoy IL-1RII expressed on human CD4$^+$ T cells by exploring its molecular mechanisms of action and analyzing its effect on Th17 responses. TCR stimulation potently induces functional IL-1RI on memory CD4$^+$ T cells and a subset of IL-1RI$^+$ CD4 T cells co-express decoy IL-1RII, conferring limited responsiveness to IL-1β. IL-RII expression is mediated through binding of a cooperative complex of the transcription factors NFAT and Foxp3 to its promoter. Differential expression of IL-1RI and IL-RII on activated CD4$^+$ T cells defines unique immunological features via modulation of IL-1 β responsiveness. Of note, de novo expression of IL-1RI is aberrantly higher in synovial CD4$^+$ T cells from RA patients, but TCR-mediated induction of IL-1RII expression is impaired compared to peripheral CD4$^+$ T cells from RA patients or healthy controls. These findings underscore the regulatory role of decoy IL-1RII on IL-1β-mediated IL-1R signaling affecting Th17 responses and contributing to pathogenesis of Th17-related disorders.

# Results

## IL-1β is a crucial for promoting Th17 responses in humans

To investigate the role of IL-1β in promoting IL-17 production in human CD4$^+$ T cells, purified memory CD4$^+$ T cells from healthy donors were stimulated with anti-CD3/CD28-coated beads with or without exogenous IL-1β. The frequency of IL-17-producing cells and their level of IL-17 production were significantly increased by exogenous IL-1β treatment (*Figure 1A and B*) in a dose-dependent manner (*Figure 1—figure supplement 1A.*). Consistent with previous reports (*Lee et al., 2012*; *Ronchi et al., 2016*; *Zielinski et al., 2012*), IL-1β also significantly enhanced the frequency of IL-17/IFN-γ double producing cells, whereas IL-10-producing cells were decreased (*Figure 1C* and *Figure 1—figure supplement 1B*). Furthermore, IL-1β treatment led to an upregulated expression of the genes which are crucial for pathogenic functions of Th17 cells such as IL-22, Casp1, and several chemokines (*Figure 1—figure supplement 1C*). As seen in *Figure 1D*, Th17-polarizing cytokines, such as IL-1β, IL-6 and IL-23, intensify the Th17 response of memory CD4$^+$ T cells as observed by the increased amount of IL-17 in the culture. This induction was substantially diminished in the absence of IL-1β. The IL-17-promoting effect of IL-1β on human memory CD4$^+$ T cells was not replaceable with IL-6 or IL-23 (*Figure 1D*), indicating the essential role of this cytokine for the Th17 response in memory CD4$^+$ T cells. In contrast to this, Th17 differentiation from naive CD4$^+$ T cells was efficiently achieved in vitro by triggering the TCR in the presence of a mixture of Th17-polarizing cytokines, although IL-17 was minimally produced in the absence of IL-1β (*Figure 1E*). Our findings demonstrate that IL-1β is important for promoting Th17 responses of human memory CD4$^+$ T cells and that IL-1β-stimulated memory CD4$^+$ T cells exhibit a major functional feature of pathogenic Th17 cells.

## TCR stimulation induces the expression of IL-1 receptors by memory CD4$^+$ T cells

The expression of receptors for IL-1β on CD4$^+$ T cells is affected by various stimuli, which may play a role in modulating responsiveness to IL-1β (*Lee et al., 2010*; *Tran et al., 2009*). We first examined the expression profile of IL-1 receptors including functional IL-1RI and the decoy IL-1RII in ex vivo and in vitro activated CD4$^+$ T cells. Approximately 10% of ex vivo CD4$^+$ T cells were found to express IL-1RI, whereas no obvious expression of IL-1RII was observed on the same cells (*Figure 2A*). These IL-1RI$^+$ memory CD4$^+$ T cells also had higher expression of IL-23R that is predominantly expressed in IL-17-producing cells (*Figure 2—figure supplement 1*; *Wilson et al., 2007*). As reported (*Lee et al., 2010*), IL-1RI-expressing CD4$^+$ T cells are predominantly memory CD45RA$^-$CD4$^+$ T cells (*Figure 2B*, p<0.001) and only a small fraction of naive CD4$^+$ T cells express IL-1RI. TCR stimulation strongly triggered expression of both IL1RI and IL-1RII by memory CD4$^+$ T cells, with rapid increase of their mRNA levels within 4 hr and sustained or even increased expression until 24 hr post-stimulation (*Figure 2—figure supplement 2A*). Of interest, the kinetics of IL-1RI and IL-1RII expression differ noticeably in that IL-1RI expression is significantly increased within 12 hr after TCR stimulation, whereas induction of IL-1RII mainly occurs in IL-1RI-expressing cells after 36 hr of stimulation (*Figure 2C–D*). To further examine the role of TCR stimulation on the expression of IL-1Rs, CD4$^+$ T cells were stimulated with different concentrations of anti-CD3 Ab for 48 hr. As seen in *Figure 2E*, IL-1RI expression was maximal even at the lowest concentration of CD3 Ab, whereas IL-1RII expression increased in a CD3 Ab concentration-dependent manner. Our data show that TCR-stimulated memory CD4$^+$ T cells can be subdivided into three subsets based on the expression of IL-1RI and IL-1RII: IL-1RI$^+$IL-1RII$^-$, IL-1RI$^+$IL-1RII$^+$, and IL-1RI$^-$IL-1RII$^-$ cells (*Figure 2C*). Around half of IL-1R$^+$ memory CD4$^+$ T cells co-expressed decoy IL-1RII on day 2 when expression of the two receptors peaked. In stark contrast with memory CD4$^+$ T cells, no expression of IL-1RI or IL-1RII by naive CD4$^+$ T cells was found to occur following TCR stimulation alone. Rather, Th17-polarizing cytokines were required for induction of IL-1RI and IL-1RII by naive CD4$^+$ T cells (*Figure 2—figure supplement 2B*). We next tested whether induced IL-1RII could functionally modulate CD4$^+$ T cell responsiveness to IL-1β. As seen in *Figure 2F*, neutralizing antibody (Ab) to IL-1RII significantly enhanced the production of IL-17 by memory CD4$^+$ T cells in response to TCR and IL-1β co-stimulation. These findings indicate that TCR triggering alone leads to induction of IL-1RI and IL-1RII by memory CD4$^+$ T cells, and both receptors are involved in regulating responsiveness to IL-1β.

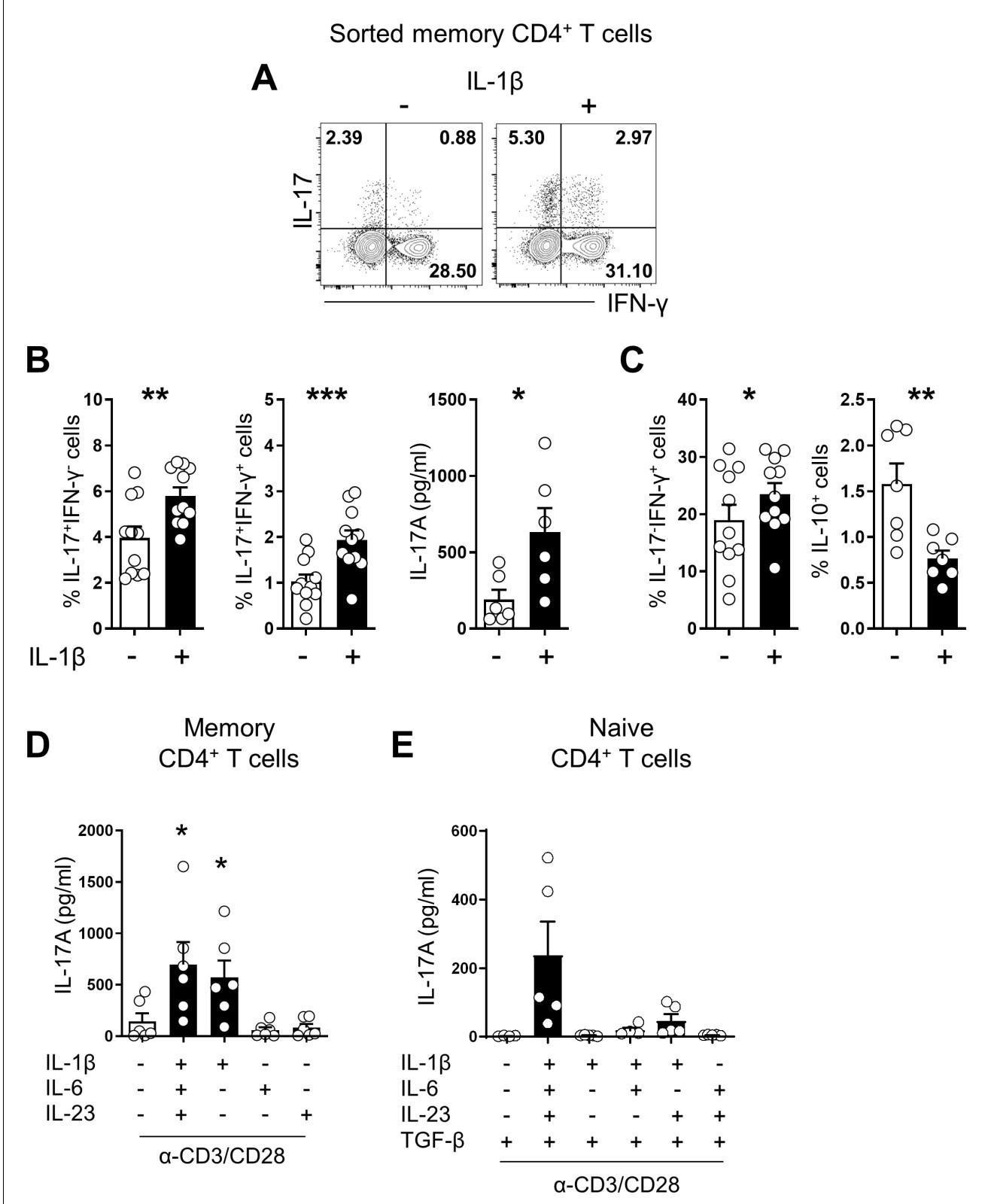

**Figure 1.** IL-1β is crucial for promoting human Th17 responses. (A) Representative flow cytometric plot of IL-17 and/or IFN-γ production by human memory CD4+ T cells obtained from HCs and stimulated for 7 days with anti-CD3/28-coated microbeads with or without rhIL-1β (5 ng/ml). (B) Frequency (%) of IL-17-producing cells and IL-17/IFN-γ double-producing cells (flow cytometry) and the amount of IL-17 in the culture supernatant of (A) (ELISA). (C) Frequency (%) of IFN-γ-producing cells and IL-10-producing cells (flow cytometry). (D) The amount of IL-17 in the culture supernatant under
*Figure 1 continued on next page*

Figure 1 continued

the indicated cytokine conditions (ELISA): rhIL-6 (25 ng/ml), rhIL-23 (25 ng/ml), and rhL-1β (5 ng/ml). (E) Purifed naive CD4[+] T cells were stimulated for 7 days with anti-CD3/28-coated microbeads in serum-free X-VIVO 10 medium under the indicated cytokine conditions. IL-17 in the culture supernatant (ELISA): rhIL-6 (25 ng/ml), rhIL-23 (25 ng/ml), rhL-1β (5 ng/ml), and rhTGF-β (10 ng/ml). Bar graphs show the mean ± SEM of six (A–D), and five (E) independent experiments. * = p<0.05, ** = p<0.01, and *** = p<0.001 by two-tailed paired t-test.

The online version of this article includes the following source data and figure supplement(s) for figure 1:

Source data 1. *Figure 1B* IL-1β significantly enhance IL-17 & IFN-γ producing memory CD4+ T cells.

Source data 2. *Figure 1C* IL-1β regulated IL-10 producing memory CD4+ T cells.

Source data 3. *Figure 1D* Th17-polarizing cytokines intensify the Th17 response of memory CD4+ T cells.

Source data 4. *Figure 1E* IL-1β is important for promoting Th17 differentiation form naïve CD4+ T cells.

Figure supplement 1. IL-1β is a crucial cytokine for promoting Th17 responses in humans.

## Differential expression of IL-1Rs defines the immunological features of CD4[+] T cells

Upregulated IL-1RII expression has been reported in several types of Tregs in humans, indicating its potential regulatory role in IL-1-dependent responses (*Tran et al., 2009*; *Ritvo et al., 2017*). Similar to a recent report (*Mercer et al., 2010*), we found that CD4[+] T cells expressing GARP, a marker of activated Tregs, exhibit preferentially increased IL-1RII expression (*Figure 3—figure supplement 1*). However, a subset of IL-1RI[+] CD4[+] T cells without GARP expression also exhibited IL-1RII expression on cells in response to TCR stimulation (*Figure 3—figure supplement 1A and B*). Thus, this suggests that IL-1RII expression is not strictly limited by activated Treg cells. To further investigate this finding, IL1RII expression was analyzed on conventional memory CD4[+] T cells from which CD25[hi]CD127[dim/-] Tregs were removed. Although CD25[hi]CD127[(dim/-)] and foxp3[+] Treg cells might be not exactly overlapped, foxp3[+] cells were markedly removed by our depletion protocol (*Figure 3—figure supplement 2*) and the residual foxp3[+] T cells mainly belonged to non-suppressive cytokine-producing Foxp3[low] T cells (III in *Figure 3—figure supplement 2C*) by the definition of the Sakaguchi group. In fact, foxp3[hi]-activated Treg cells (II in *Figure 3—figure supplement 2C*) were mostly removed by our Treg-depletion protocol. TCR triggering led to the induction of IL-1RII expression even by Treg-depleted T cells (*Figure 3A*). Of note, these IL-1RII[+] memory CD4[+] T cells have higher expression of Foxp3 that could be induced in Treg-depleted conventional CD4[+] T cells with Smad3- and NFAT-dependent manner (*Figure 3—figure supplement 3*) and other Treg-related markers such as CD39, CD73, and CTLA-4 (*Figure 3B*) compared with the other two subsets. To characterize the features of IL-1RI[+]IL-1RII[+] memory CD4[+] T cells, chemokine receptors and cytokine expression was analyzed in three populations (IL-1RI[+]IL-1RII[-], IL-1RI[+]IL-1RII[+], and IL-1RI[-]IL-1RII[-] cells). As seen in *Figure 3C*, around 50% of IL-1RI[+]IL-1RII[-] and IL-1RI[+]IL-1RII[+] had a Th17-associated phenotype (CD161[+]CCR6[+]), whereas IL-1RI[-]IL-1RII[-] cells mainly consist of Th1 (CD161[-]CXCR3[+]CCR6[-]) and non-Th1/17 subsets. Moreover, a higher frequency of IL-1RI[+]IL-1RII[+] cells showed a phenotype typical of ex-Th17 cells (CD161[+]CXCR3[+]CCR6[+]), whereas the IL-1RI[+]IL-1RII[-] subset included more cells with the Th17 phenotype (CD161[+]CXCR3[-]CCR6[+])(*Basdeo et al., 2017*). Although the frequency of IL-23R was comparable among three populations, its expression level was significantly higher in IL-1RI[+]IL-1RII[-] cells than IL-1RI[+]IL-1RII[+], and IL-1RI[-]IL-1RII[-] cells (*Figure 3—figure supplement 4*). Intracellular cytokine staining (ICS) data revealed that IL-1RI[+]IL-1RII[-] and IL-1RI[+]IL-1RII[+] cells had different cytokine production profiles, showing that IL-1RI[+]IL-1RII[+] consists of fewer IL-17-producing, but more IFN-γ-producing, cells than IL-1RI[+]IL-1RII[-] cells (*Figure 3D*). Our findings demonstrate that IL-1RII is inducible by activated memory CD4[+] T cells, associated with Foxp3 expression, and IL-1RII[+] CD4[+] T cells have unique immunological features.

## NFAT and Foxp3 are essential for IL-1RII expression in memory CD4[+] T cells

Since TCR-induced Foxp3[+] T cells preferentially express IL-1RII (*Figure 3A*), we hypothesized that Treg-related transcriptional factors (TFs) are involved in *IL-1RII* gene expression. In preliminary experiments using inhibitors for TFs related to Treg cells, we found that cyclosporin A (CsA), an inhibitor of calcineurin, selectively represses the frequency of IL-1RII[+] cells among TCR-induced IL-1RI[+] memory CD4[+] T cells in a dose-dependent manner (*Figure 4A and B*). The immunosuppressive

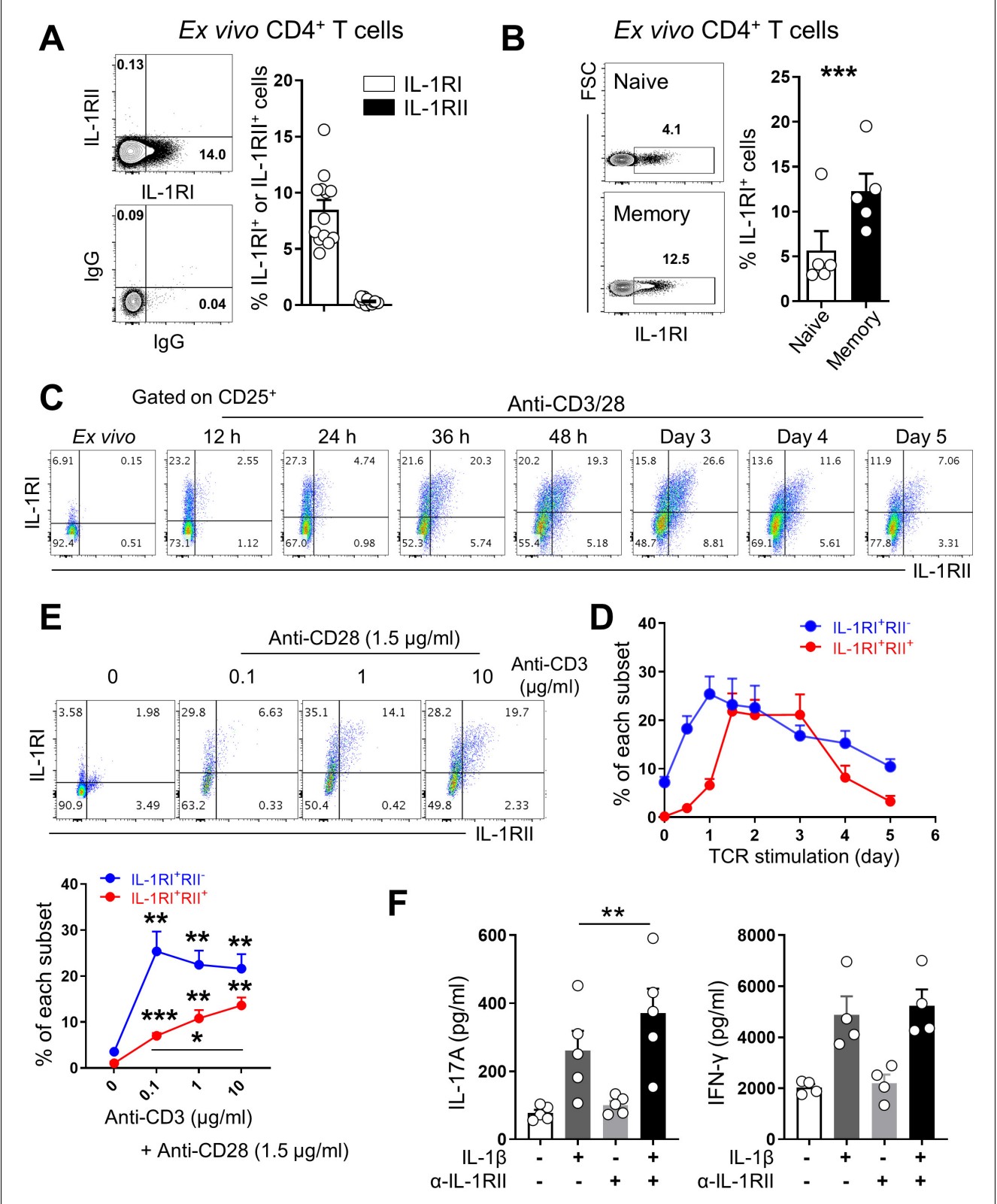

**Figure 2.** Dynamic regulation of IL-1β receptors by memory CD4[+] T cells upon TCR stimulation. (**A**) Representative flow cytometric plot and the frequency (%) of functional IL-1RI- and decoy IL-1RII-expressing CD4[+] T cells of HCs (n = 13). (**B**) Representative flow cytometric plot and the frequency (%) of IL-1RI[+] cells in naive (CD45RA[+]CCR7[+]) or memory (CD45RA[−]) CD4[+] T-cell subset of HCs (n = 13). (**C**) Representative flow cytometric plot of change in IL-1RI and IL-1RII expression by TCR-stimulated memory CD4[+] T cells (n = 7). (**D**) Time kinetics of IL-1RI and IL-1RII expression on TCR-
*Figure 2 continued on next page*

*Figure 2 continued*

stimulated memory CD4+ T cells (n = 5). (**E**) Representative flow cytometric plot and the frequency (%) of IL-1RI and IL-1RII expression by memory CD4+ T cells in response to stimulation with different concentrations of anti-CD3 and 1.5 µg/ml of anti-CD28 Ab for 48 hr (n = 5). (**F**) The amounts of IL-17A (left) and IFN-γ (right) in culture supernatants of memory CD4+ T cells stimulated with anti-CD3/28-coated microbeads for 7 days with or without IL-1β (n = 5). Anti-IL-1RII neutralizing Ab or control isotype Ab was added into the culture at day 2 post-stimulation. Bar graphs and line graphs show the mean ± SEM. * = p<0.05, ** = p<0.01, and *** = p<0.001 by two-tailed paired *t*-test.

The online version of this article includes the following source data and figure supplement(s) for figure 2:

Source data 1. *Figure 2A* Ex vivo expression of IL-1RI & IL-1RII on CD4+ T cells.
Source data 2. *Figure 2B* Ex vivo expression of IL-1RI between naïve and memory CD4+ T cells.
Source data 3. *Figure 2B* Ex vivo expression of IL-1RI between naïve and memory CD4+ T cells.
Source data 4. *Figure 2D* Time kinetics of IL-1RI & IL-1RII.
Source data 5. *Figure 2E* Effect of TCR signaling strength on expression of IL-1RI & IL-1RII.
Figure supplement 1. The differential expression of IL-23R on IL-1RI+ and IL-1RI- memory CD4+ T cells in humans.
Figure supplement 2. The expression of receptors for IL-1β is dynamically changed by various stimulations of CD4+ T cells.

effect of CsA is mediated by inhibiting calcineurin-mediated dephosphorylation of the nuclear factor of activated T-cells (NFATc), which plays a critical role in peripheral activation and differentiation of Tregs (*Kiani et al., 2000*). To further investigate whether NFAT is directly involved in *IL-1RII* gene expression, purified memory CD4+ T cells were treated with cell-permeable peptide (CPP)-conjugated VIVIT, a selective and potent inhibitor of calcineurin/NFAT interaction, and stimulated with anti-CD3/CD28 mAbs-coated beads for 24 hr. As seen in *Figure 4C and D*, expression of IL-1RII mRNA was completely abrogated and its protein level was also significantly diminished. We next investigated if Foxp3, a master TF in Tregs, is required for *IL-1RII* gene expression. Consistent with a previous report (*Kang et al., 2012*), 1,25-Dihydroxyvitamin D3 [1,25(OH)$_2$VD$_3$] promoted Foxp3 expression in memory CD4+ T cells following TCR stimulation in the presence of IL-2, and induction of Foxp3 was clearly attenuated by CsA treatment (*Figure 4E*). Moreover, the expression of IL-1RII, but not IL-1RI, was found to be significantly upregulated by treatment with 1,25(OH)$_2$VD$_3$ and IL-2 in memory CD4+ T cells at day 2 post-stimulation and resultantly, the ratio of IL-1RII+ to IL-1RI+ increased two fold (*Figure 4F–H*). This suggests that Foxp3 upregulation is associated with induction of IL-1RII. Furthermore, our findings demonstrate that NFAT and Foxp3 are essential for expression of IL-1RII by memory CD4+ T cells in humans.

## NFAT/FOXP3 interaction is responsible for expression of IL-1RII by human memory CD4+ T cells

It has been demonstrated that Treg activity is modulated by a cooperative complex of NFAT and FOXP3[27]. Since both TFs are critical for IL-1RII expression (*Figure 4*), we next examined whether formation of a cooperative NFAT/FOXP3 complex is responsible for upregulation of IL-1RII expression on memory CD4+ T cells upon TCR stimulation. To this end, peptide FOXP3 393–403, a specific inhibitor of the NFAT/FOXP3 interaction, was used to treat TCR-stimulated memory CD4+ T cells for 48 hr. Although this peptide inhibitor was initially designed for murine Tregs, this same inhibitory effect was verified for the human NFAT/FOXP3 interaction through inhibition of CD25 induction, a target of this complex (*Figure 5A*), as described in a previous report (*Lozano et al., 2015*). Our data show that peptide FOXP3 393–403 significantly inhibits IL-1RII expression by approximately 50% compared with the control peptide, FOXP3 399A, (*Figure 5B and C*). Thus, this confirms the important role of the NFAT/FOXP3 complex in IL-1RII expression.

To determine whether the NFAT/FOXP3 complex directly binds to the *IL-1RII* gene, in silico screening was conducted for potential NFAT and FOXP3-binding sites in the minimal promoter of the human *IL-1RII* gene using PROMO 3.0 software, an algorithm that predicts nuclear-receptor-binding elements (*Messeguer et al., 2002*). Potential response elements for NFAT and FOXP3 were closely located in the conserved non-coding sequence (CNS) region (−1309 to −1203) of the *IL-1RII* gene (*Figure 5—figure supplement 1*), implying the importance of this region for IL-1RII expression. Three luciferase reporter constructs (*Figure 5D*) containing 1.3 kb (pGL4/IL-1RII-a1414), 0.7 kb (pGL4/IL-1RII-a814), and 0.1 kb (pGL4/IL-1RII-a215) of the *IL-1RII* promoter were transduced into the Jurkat leukemic T-cell line. Due to a lack of Foxp3 expression in conventional Jurkat cells, we generated a Foxp3-expressing Jurkat cell line (Foxp3-Jurkat) using lentiviral vector transduction

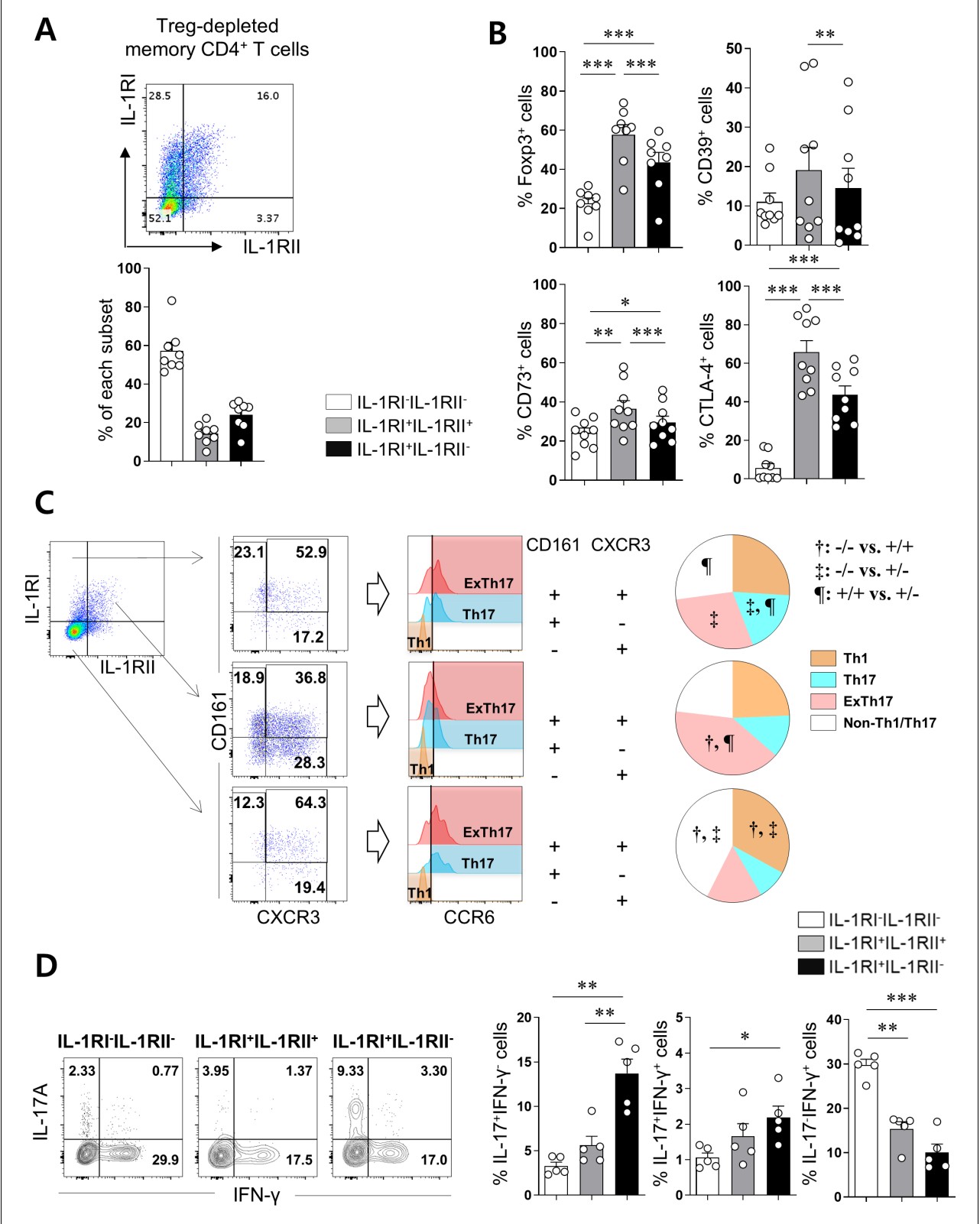

**Figure 3.** Differential expression pattern of IL-1Rs defines immunological features of CD4+ T cells. (A) Treg-depleted memory CD4+ T cells were stimulated with anti-CD3/28-coated microbeads for 48 hr. The expression of IL-1RI and IL-1RII were analyzed by flow cytometry (n = 12). (B) The frequency (%) of Treg-related marker-expressing cells in IL-1RI+IL-1RII-, IL-1RI+IL-1RII+, and IL-1RI-IL-1RII- memory CD4+ T cell populations at day 2 post-stimulation (n = 12). (C) Representative flow cytometric plot and the frequency (%) of Th1 (CCR6-CD161-CXCR3+), Th17 (CCR6+CD161+CXCR3-), and ex-

*Figure 3 continued on next page*

Figure 3 continued

Th17 (CCR6$^+$CD161$^+$CXCR3$^+$) cells in IL-1RI$^+$IL-1RII$^-$, IL-1RI$^+$IL-1RII$^+$, and IL-1RI$^-$IL-1RII$^-$ memory CD4$^+$ T cell populations (n = 9). (D) Intracellular cytokine staining (ICS) of IL-17- and IFN-γ in IL-1RI$^+$IL-1RII$^-$, IL-1RI$^+$IL-1RII$^+$, and IL-1RI$^-$IL-1RII$^-$ memory CD4$^+$ T cell populations (n = 4). Bar graphs and pie charts show the mean ± SEM. * = p<0.05, ** = p<0.01, and *** = p<0.001 by two-tailed paired t-test. †, ‡, and ¶ = p<0.05: compared between indicated subsets by two-tailed paired t-test (C).

The online version of this article includes the following source data and figure supplement(s) for figure 3:

**Source data 1.** *Figure 3A* Frequency of IL-1RI+IL-1RII-, IL-1RI+IL-1RII+, and IL-1RI-IL-1RII- subset of Treg-depleted memory CD4+ T cells.

**Source data 2.** *Figure 3B* Expression of Treg related markers on L-1RI+IL-1RII-, IL-1RI+IL-1RII+, and IL-1RI-IL-1RII- subsets.

**Source data 3.** *Figure 3C* Frequency of ex-Th17, Th17, and Th1 of L-1RI+IL-1RII-, IL-1RI+IL-1RII+, and IL-1RI-IL-1RII- subsets.

**Source data 4.** *Figure 3D* Expression of IL-17 & IFN-γ in L-1RI+IL-1RII-, IL-1RI+IL-1RII+, and IL-1RI-IL-1RII- subsets.

**Figure supplement 1.** CD4$^+$ T cells expressing GARP, a marker of activated Tregs, preferentially, but not exclusively, increase IL-1RII expression.

**Figure supplement 2.** Efficiency of sorting-based Treg depletion from CD4$^+$ T cells.

**Figure supplement 3.** Critical roles of Smad3 and NFAT for Foxp3 expression in non-Treg memory CD4$^+$ T cells.

**Figure supplement 4.** The correlation of IL-23R with IL-1RI$^+$IL-1RII$^+$ and IL-1RI$^+$IL-1RII$^-$ subset of memory CD4$^+$ T cells.

(*Figure 5—figure supplement 2A*). IL-1RII expression was markedly enhanced in Foxp3-Jurkat cells following stimulation with PMA and ionomycin (*Figure 5—figure supplement 2B and C*), which are known to cause NFAT activation (*Brignall et al., 2017*). As seen in *Figure 5D*, Foxp3-Jurkat cells transfected with pGL4/IL-1RII-a1414 including the CNS region exhibited significantly higher luciferase activity compared with the other two constructs. We determined the functional relevance of these TF binding sites in the CNS region of the *IL-1RII* promoter by mutational analysis. Mutation of three bases within each NFAT binding site in the CNS region resulted in a significant decrease in IL-1RII promoter activity (*Figure 5D*), indicating the importance of these sites for promoter function.

To further investigate whether NFAT and Foxp3 bind the predicted binding sites in the *IL-1RII* promoter region, we performed a CHIP assay with TCR-stimulated memory CD4$^+$ T cells. NFAT recruitment to the DNA response element (NFAT B1: −1209 to −1188) in the CNS region of IL-1RII promoter was significantly enriched upon TCR stimulation, whereas Foxp3 was observed to bind a slight distant from the CNS region (Foxp3 B2: −425 to −418) of IL-1RII promoter (*Figure 5E and F*). To further confirm this finding, NFAT recruitment to the B1 site were evaluated in Jurkat cells with or without foxp3 expression. NFAT recruitment to the NFAT BC (binding control), a binding site in human *IL-2* promoter region, were comparable between Jurkat cells with or without foxp3 expression, whereas NFAT recruitment to the B1 site were significantly enriched in Foxp3-expressing Jurkat cells upon stimulation with PMA and ionomycin compared with Jurkat cells without foxp3 expression (*Figure 5—figure supplement 3*). Collectively, these results suggest that NFAT and Foxp3 bind directly to the IL-1RII promoter and regulate its transcription.

## IL-1RII ameliorates IL-1β-mediated Th17 responses and confers features of Tregs to IL-1RI$^+$ memory CD4$^+$ T cells

Our data thus far demonstrate that the induction of IL-1RII on IL-1RI$^+$ memory CD4$^+$ T cells is controlled by NFAT/FOXP3, which plays an important role in the induction of Tregs in humans, and this induction leads to weakened IL-1β-mediated Th17 responses. Accumulating evidence demonstrates that Th17 and Treg cells display instability and plasticity and can interconvert under specific conditions to modulate immune responses (*Gagliani et al., 2015*; *Obermajer et al., 2014*). Therefore, we next investigated how IL-1RII expression influences the balance between Th17 and Treg cells. To address this question, Treg-depleted memory CD4$^+$ T cells were stimulated for 48 hr to induce the expression of IL-1RI and IL-1RII. Three different subsets, IL-1RI$^+$IL-1RII$^-$, IL-1RI$^+$IL-1RII$^+$, and IL-1RI$^-$IL-1RII$^-$ cells, were purified by cell sorting, and stimulated through their TCR for 5 days in the presence of IL-1β. Following exposure to IL-1β, IL-1RI$^+$ cells were the major IL-17 producing population regardless of IL-1RII expression, whereas IL-1RI$^-$IL-1RII$^-$ cells predominantly produced IFN-γ (*Figure 6A and B*). IL-1RII expression, however, attenuated the Th17 response of IL-1RI$^+$ cells, as shown by a reduced frequency of IL-17-producing cells, and in particular IL-17 and IFN-γ dual producing cells known as typical pathogenic Th17 cells (*Figure 6A*). Our qPCR assay showed several pathogenic Th17 cell-associated genes including IL-22, CCL3, and CSF2 were upregulated in IL-1RI$^+$IL-1RII$^-$ cells than IL-1RI$^+$ IL-1RII$^+$ cells (*Figure 6—figure supplement 1*). In contrast, increased expression of Foxp3 was observed in IL-17-producing IL-1RI$^+$RII$^+$ compared with IL-1RI$^+$ILRI$^-$ cells,

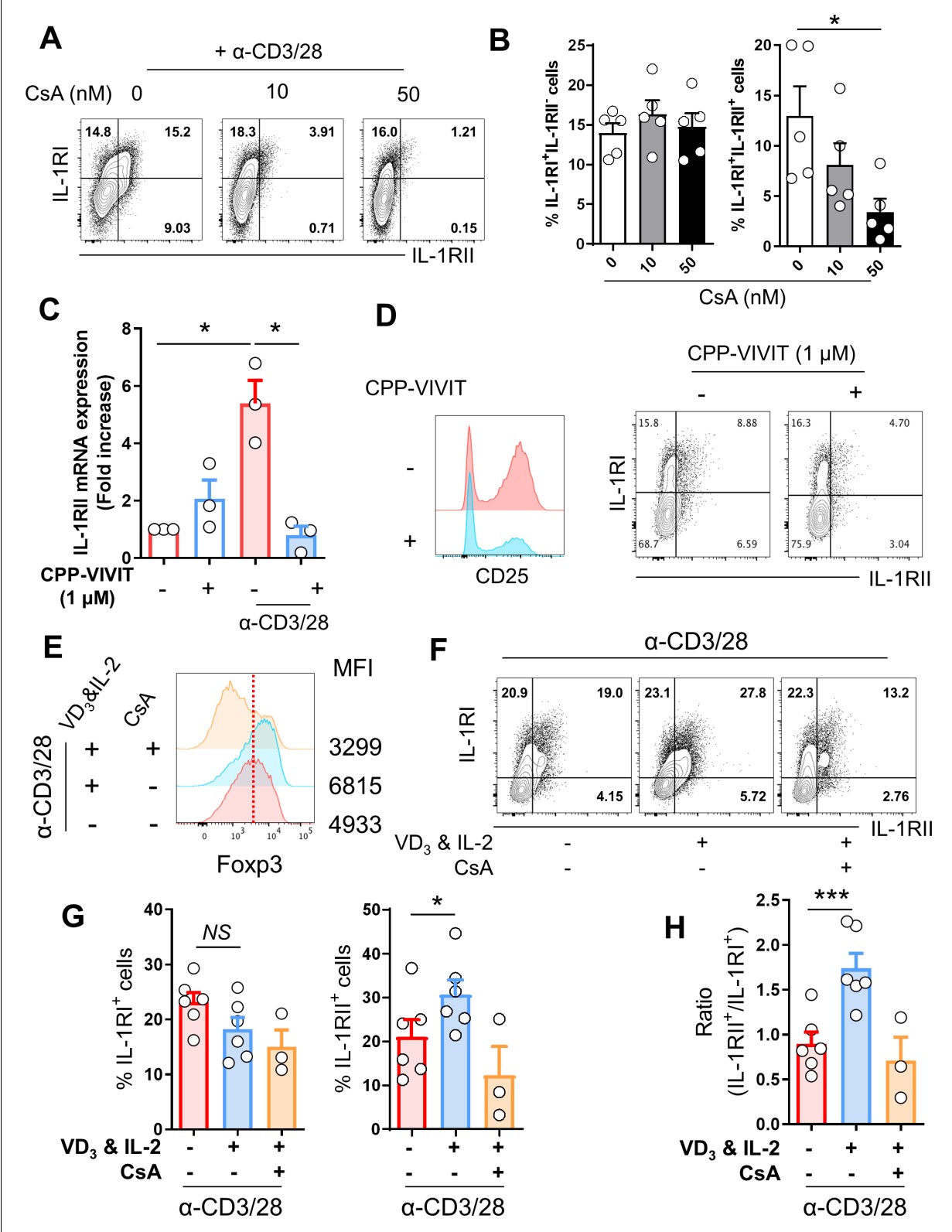

**Figure 4.** NFAT and Foxp3 are essential for IL-1RII expression in memory CD4+ T cells. (**A and B**) Representative flow cytometric plot (**A**) and the frequency (**B**) of IL-1RI and IL-1RII expression on TCR-stimulated memory CD4+ T cells treated with the indicated concentration of CsA, a chemical inhibitor of NFAT at day 2 post-stimulation (n = 5). (**C**) The expression of IL-1RII mRNA in TCR-stimulated or unstimulated memory CD4+ T cells in the presence or absence of CPP-VIVIT (1 μM), NFAT-specific peptide inhibitor (n = 4). (**D**) The expression of CD25, IL-1RI, and IL-1RII on TCR-stimulated

*Figure 4 continued on next page*

*Figure 4 continued*

memory CD4 T cells in the presence or absence of CPP-VIVIT (n = 3). (**E**) Representative histogram plot of Foxp3 in TCR-stimulated memory CD4$^+$ T cells in the presence or absence of 1,25-dihydroxyvitamin D3 (VD$_3$; 10 μM) and rhIL-2 (250 IU/ml) with or without CsA. MFI indicates mean fluorescent intensity. (**F**) Flow cytometric analysis of IL-1RI and IL-1RII on memory CD4$^+$ T cell under same conditions as in (**E**). (**G and H**) The frequencies (%) of IL-1RI$^+$ and IL-1RII$^+$ cells and the ratio of IL-1RII$^+$ to IL-1RI$^+$ cells under same conditions in (**E**) (n = 6). Bar graphs show the mean ± SEM. * = p<0.05, ** = p<0.01, and *** = p<0.001 by two-tailed paired *t*-test.

The online version of this article includes the following source data for figure 4:

**Source data 1.** *Figure 4A* NFAT inhibitor CsA selectivley repress the expression of IL-1RII.
**Source data 2.** *Figure 4B* NFAT inhibiton peptide VIVIT repress the expression of IL-1RII.
**Source data 3.** *Figure 4G* expression of IL-1RII significantly upregulated by treatment with 1,25(OH)2VD3 and IL-2 in memory CD4+ T cells.
**Source data 4.** *Figure 4H* Ratio of IL-1RII+/IL-1RI+.

implying that IL-1RII is associated with modulating the balance between Th17 cells and Tregs (*Figure 6C*). Confirmatory FACS analysis revealed that CTLA -4 expression on IL-1RI$^+$IL-1RII$^+$ cells is significantly higher than on IL-1RI$^+$IL-1RII$^-$, whereas IL-1RI$^+$IL-1RII$^-$ cells had higher frequencies of cells expressing CCR6 and CD161, Th17-related markers, than did IL-1RI$^+$IL-1RII$^+$ cells (*Figure 6D*). Although the receptor of IL-23, which is a critical cytokine for the differentiation, commitment, and survival of Th17 cells, was upregulated in IL-1RI$^+$IL-1RII$^-$ cells (*Figure 3—figure supplement 4*), the increase in IL-17 production by IL-23 was minimal in the in vitro system. This may be because IL-23 receptor expression on IL-1RI$^+$ cells is limited and the induction kinetic of IL-23R is different from IL-1Rs (*Figure 6—figure supplement 2*). These data demonstrate that the expression of IL-1RII plays a critical role in maintaining the balance between Th17 and Treg cells in humans.

## Aberrant expression of IL-1Rs in synovial CD4$^+$ T cells in patients with rheumatoid arthritis (RA)

It has been reported that the frequency of Foxp3$^+$CD4$^+$ T cells is markedly increased in the synovial cavity of RA patients and these cells are able to produce IL-17 (*Komatsu et al., 2014*; *Du et al., 2014*; *Afzali et al., 2013*). Furthermore, IL-1β is obviously elevated in synovial fluid (SF) of RA patients (*Kim et al., 2016*). This finding prompted us to examine the expression profile of IL-1RI and IL-1RII in synovial and peripheral CD4$^+$ T cells in RA patients. The frequency of IL-1RI-expressing cells was significantly higher in ex vivo memory CD4$^+$ T cells derived from synovial fluid (16.35 ± 1.70%) and peripheral blood (14.49 ± 1.20%) of RA patients compared with counterpart cells (7.11 ± 0.85%; p<0.001) from age-match healthy controls (HCs) (*Figure 7A and B*). Like its expression on peripheral CD4$^+$ T cells of HCs, IL-1RII was minimally expressed by ex vivo CD4$^+$ T cells of RA patients, with the exception of a small population of synovial CD4$^+$ T cells that expressed higher levels of IL-1RII, indicating their activated state. Of importance, in IL-1RI$^+$ cells TCR-mediated induction of IL-1RII was significantly impaired in synovial CD4$^+$ T cells compared with peripheral CD4$^+$ T cells of HCs and even of RA patients (*Figure 7C and D*). As a consequence, the ratio of IL-1RI$^+$IL-1RII$^+$ to IL-1RI$^+$IL-1RII$^-$ was lower in RA patients than HCs (*Figure 7E*). Given that the frequency of foxp3$^+$ CD4$^+$ T cells in PBMC of HC was comparable to that in RA PBMC and was even significant lower than that in RA SFMC, impaired expression of IL-1RII in RA memory CD4$^+$ T cells was not likely to be caused by the reduced frequency of Foxp3$^+$ Treg cells (*Figure 7—figure supplement 1*). As seen in *Figure 7F*, the reduced induction of IL-1RII on TCR-stimulated memory CD4$^+$ T cells in RA patients is likely linked to augmented induction (3.3 fold-increase) of IL-1β-mediated IL-17 production in response to TCR stimulation compared with HCs (2.4 fold-increase) (*Figure 7F*) because this induction of RA patients was observed at a comparable level to that of CD4$^+$ T cells of HCs with anti-IL-1RII neutralizing Ab (*Figure 7F and G*). This data suggests that TCR-induced IL-1RII is associated with modulation of IL-17 production in human CD4$^+$ T cells.

## Discussion

Th17 cells are a unique subset of CD4$^+$ effector T cells that elicit protective immune responses against extracellular bacterial and fungal pathogens (*Gaffen, 2009*). Th17 cells have been also implicated in the pathogenesis of many autoimmune disorders (*Lee et al., 2012*; *Tabarkiewicz et al.,*

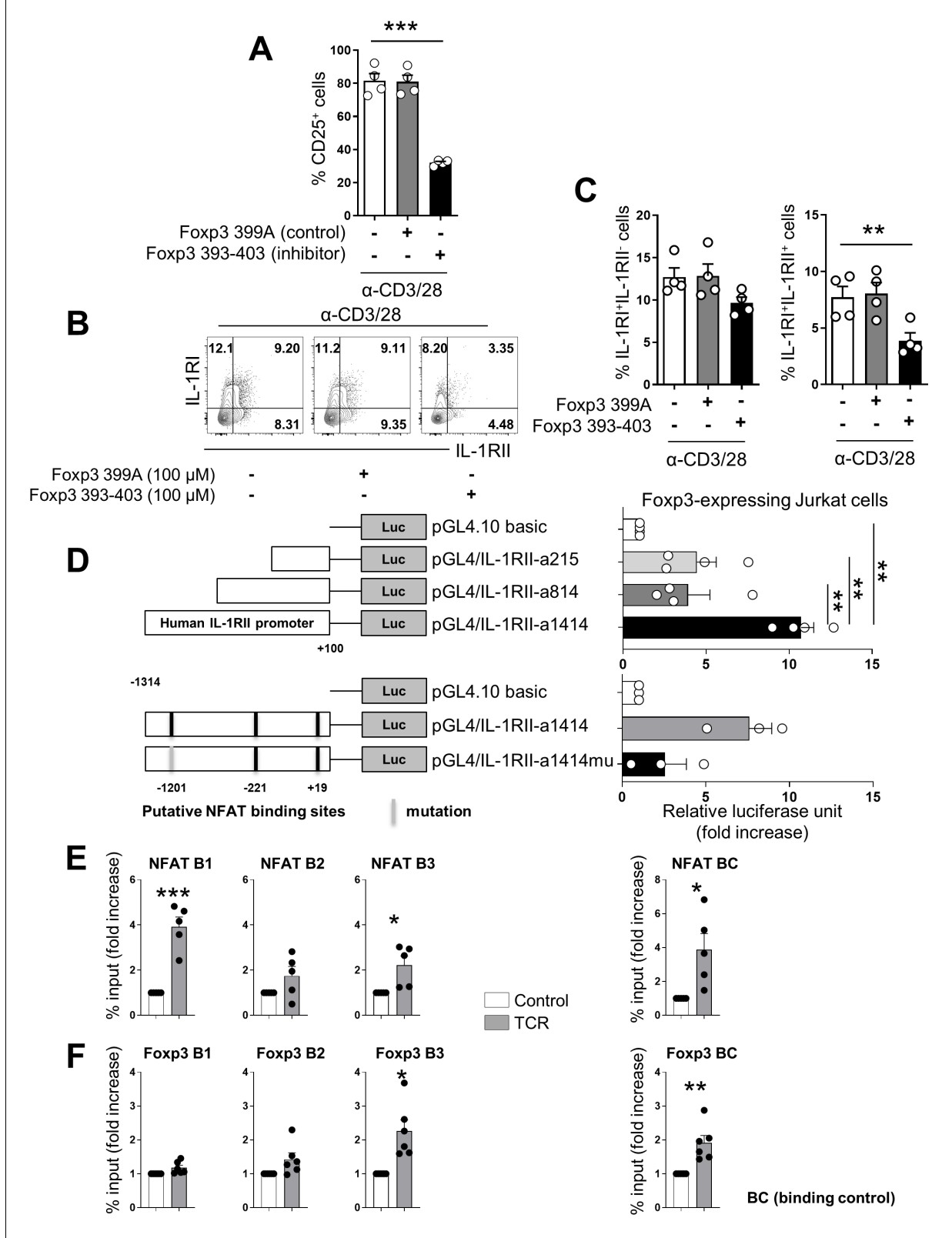

**Figure 5.** NFAT/FOXP3 interaction is responsible for expression of IL-1RII by human memory CD4[+] T cells. (**A**) The frequency (%) of CD25, a NFAT-dependent molecule, on TCR-stimulated memory CD4[+] T cells in the presence of Foxp3 393–403 peptide (100 μM), NFAT/FOXP3 interaction inhibition peptide, or control Foxp3 399 peptide (100 μM) at day 2 post-stimulation (n = 4). (**B and C**) Representative flow cytometric plot (**B**) and the frequencies (**C**) of IL-1RI and IL-1RII expression on TCR-stimulated memory CD4[+] T cells in the presence of Foxp3 393–403 peptide or control Foxp3 399 peptide at

*Figure 5 continued on next page*

Figure 5 continued

day 2 post-stimulation (n = 4). (D) Foxp3-expressing Jurkat cells were transfected with the indicated pGL4.10 luciferase vectors (pGL4/IL-1RII-a1414, pGL4/IL-1RII-a814, and pGL4/IL-1RII-a215) and pGL4.74 control renilla vector as internal control, followed by stimulation with PMA and ionomycin for 48 hr (upper panel: n = 4). Comparison of luciferase activity between pGL4/IL-1RII-a1414 and pGL4/IL-1RII-a1414mu (−1209 to −1188 NFAT-binding motif mutant) transfected Foxp3-expressing Jurkat cells (lower panel: n = 4). Luciferase activity was determinded using dual lucifease assay system. (E and F) Purified memory CD4$^+$ T cells were stimulated with plate-bound anti-CD3/28 mAbs for 24 hr. ChIP qPCR was perfomed using anti-NFATc2 or anti-Foxp3 Ab at the *IL-1RII* promoter region. Enrichment of NFAT within four putative NFAT binding motifs (E) and enrichment of Foxp3 within three putative Foxp3-binding motifs (F) was analyzed by qPCR (n = 5 or 6). There are two (−1209 ~ −1188 of *IL-1RII* promoter), one (−649 ~ −639), and one (+10 ~ +18) binding sites for NFAT B1, B2 and B3, respectively. NFAT BC (binding control) has a binding site in human *IL-2* promoter region. There are three (−4652 ~ −4611), one (−1236 ~ −1230), and one (−425 ~ −418) binding sites for Foxp3 B1, B2 and B3, respectively. Foxp3 BC (binding control) has a binding site in the human *IL-2Rα* promoter region (*Zhang et al., 2013*). Bar graphs show the mean ± SEM. * = p<0.05, ** = p<0.01, and *** = p<0.001 by two-tailed paired *t*-test.

The online version of this article includes the following source data and figure supplement(s) for figure 5:

Source data 1. *Figure 5A* The inhibitory effect of FOXP3 393–403, a specific inhibitor of the NFAT/FOXP3 interaction.

Source data 2. *Figure 5C* NFAT/Foxp3 interaction inhibitor significantly repress the IL-1RII expression.

Source data 3. *Figure 5D* IL-1RII promoer activity measured via luciferase assay.

Source data 4. *Figure 5E and F* Result of NFAT & Foxp3 ChIP-qPCR via IL-1RII promter.

Figure supplement 1. Analysis of potential binding sites of NFAT and Foxp3 in the minimal promoter of the human *IL-1RII* gene and schematic diagram of three luciferase reporter constructs.

Figure supplement 2. Overexpression of Foxp3 in Jurkat cells leads to an increase in the expression of IL-1RII.

Figure supplement 3. Cooperation of NFAT/Foxp3 via binding of NFAT on NFAT B1 site.

---

*2015*). Unlike conventional Th1 and Th2 cells, Th17 cells exhibit a substantial degree of instability, heterogeneity, and plasticity (*Stadhouders et al., 2018*). Many molecular mediators are closely involved in regulating plasticity of Th17 cells (*Lee et al., 2012*; *Wang et al., 2015*; *Ichiyama et al., 2016*; *Stockinger and Omenetti, 2017*). Among these, cytokines are critical determinants in the fate of Th17 cells. TGF-β and IL-6 give rise to nonpathogenic or beneficial Th17 cells, whereas IL-1β, IL-6 and IL-23 are involved in generation of pathogenic Th17 cells (*Lee et al., 2012*). Further, cell-fate mapping mouse models demonstrate that pathogenic Th17 cells can further convert into Th1 cells producing IFN-γ, but not IL-17 (ex-Th17 Th1 cells), in an IL-23-dependent manner (*Hirota et al., 2011*), whereas TGF-β is critical for transdifferentiation of Th17 cells into Treg cells (ex-Th17 Treg cells) during the resolution of inflammation or in the cancer microenvironment (*Gagliani et al., 2015*; *Downs-Canner et al., 2017*).

IL-1β is an essential cytokine for Th17 biology in humans that boosts Th17 production, induces pathogenic Th17 cells, and impairs the Th17-Treg balance (*Basu et al., 2015*). Activities of IL-1 are tightly controlled at different levels due to its potent proinflammatory effects. IL-1 receptor antagonist (IL-1Ra) and IL-1RII are negative regulators of the IL-1 system. IL-1Ra binds functional IL-1RI with no recruitment of IL-1RAcP, resulting in the repression of IL-1 binding in a competitive manner, whereas the decoy receptor, IL-1RII cannot transduce the signal initialized by the IL-1RI and IL-1Ra complex (*Schlüter et al., 2018*). We previously reported that the expression of IL-1RI and IL-1RII on CD4$^+$ T cells is dynamically regulated in response to environmental factors such as TCR or cytokine stimulation, and such regulation influences Th17 cell responses (*Lee et al., 2010*). Furthermore, IL-1RII mRNA is expressed at a higher level in IL-1RI$^+$CD4$^+$ T cells than in IL-1RI$^-$CD4$^+$ T cells and blockade by IL-1RII leads to reduced IL-17 production (*Lee et al., 2010*). However, the mechanism modulating IL-1RII expression and its role during the immune response remains unclear.

The expression of IL-1 and IL-1RII is tightly restricted under steady-state conditions, but is upregulated during inflammation (*Schlüter et al., 2018*). In contrast to IL-1RI, which is ubiquitously expressed at least at low levels, IL-1RII expression is confined to certain cell types, including monocytes, macrophages, some T cells in humans and microglial cells, osteoclasts and neutrophils in mice. Early studies showed that anti-inflammatory signals such as IL-4 and IL-13, as well as dexamethasone stimulate IL-1RII expression by monocytes and macrophages, suggesting its contribution to the anti-inflammatory effect (*Colotta et al., 1996*; *Colotta et al., 1993*). IL-1RII can be co-expressed with IL-1RI on the cell surface, albeit at a low copy number, and counterpoises the system by secluding IL-1 and IL-1RAcP from the neighboring signaling receptor IL-1RI (*Colotta et al., 1993*; *Neumann et al., 2000*). A recent study demonstrated that the cytosolic form of IL-1RII, expressed in certain types of cells, interacts with pro-IL-1α preventing cleavage and its calpain-dependent

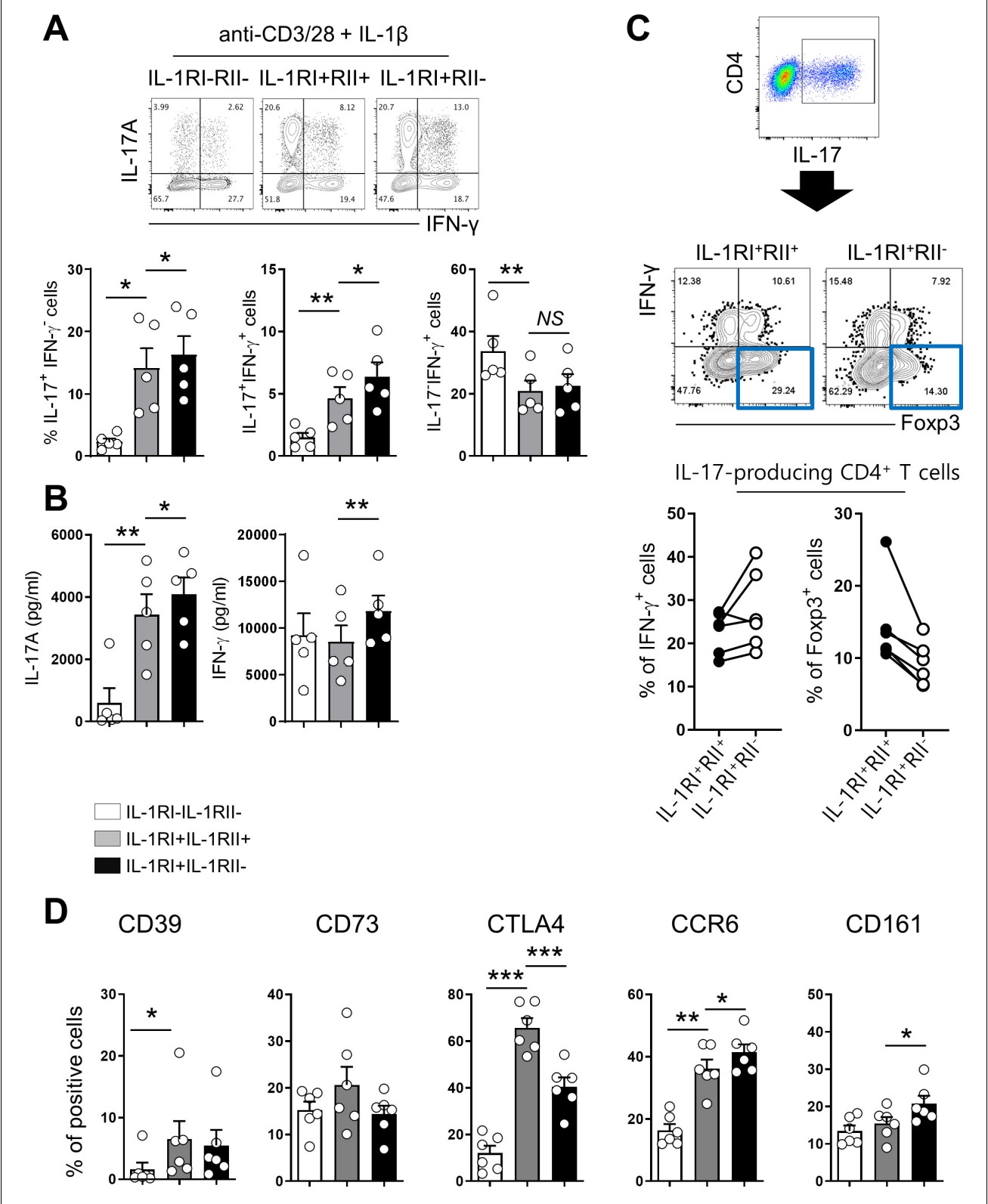

**Figure 6.** Differential expression of IL-1RI and IL-1RII on activated CD4[+] T cells defines unique immunological features of cells. Treg-depleted memory CD4[+] T cells were stimulated for 48 hr with anti-CD3/28-coated microbeads to induce IL-1RI and IL-1RII expression. Cells were sorted into IL-1RI[+]IL-1RII[-], IL-1RI[+]IL-1RII[+], and IL-1RI[-]IL-1RII[-] cells and cultured for another 5 days with rhIL-1β (5 ng/ml) and rhIL-2 (50 IU/ml). (**A**) Representative flow cytometric plot and the frequencies of IL-17- and IFN-γ-producing cells in sorted IL-1RI[+]IL-1RII[-], IL-1RI[+]IL-1RII[+], and IL-1RI[-]IL-1RII[-] memory CD4[+] T cells

*Figure 6 continued on next page*

*Figure 6 continued*

at day 7 post-stimulation (n = 6). (**B**) The amount of IL-17 and IFN-γ in the culture supernatant (**A**) by ELISA (n = 6). (**C**) Representative flow cytometric plot and the frequency of IFN-γ⁺ cells and Foxp3⁺ cells in the IL-17 producing IL-1RI⁺IL-1RII- or IL-1RI⁺IL-1RII⁺ cells (n = 5). (**D**) The frequencies of Treg-related marker- or Th17-related marker-expressing cells on sorted IL-1RI⁺IL-1RII⁻, IL-1RI⁺IL-1RII⁺, and IL-1RI⁻IL-1RII⁻ memory CD4⁺ T cells at day 7 post-stimulation (n = 4). Bar graphs show the mean ± SEM. * = p<0.05, ** = p<0.01, and *** = p<0.001 by two-tailed paired *t*-test.

The online version of this article includes the following source data and figure supplement(s) for figure 6:

**Source data 1.** *Figure 6A* Frequency of IL-17 & IFN-γ producing sorted IL-1RI+IL-1RII-, IL-1RI+IL-1RII+, and IL-1RI-IL-1RII- cells.
**Source data 2.** *Figure 6B* Concentraion of IL-17 & IFN-γ in the culture supernatant of sorted IL-1RI+IL-1RII-, IL-1RI+IL-1RII+, and IL-1RI-IL-1RII- cells.
**Source data 3.** *Figure 6C* Frequency of Foxp3 & IFN-γ producing cells of IL-17 producing IL-1RI+IL-1RII- and IL-1RI+IL-1RII+ cells.
**Source data 4.** *Figure 6D* Expression of Treg related markers on sorted L-1RI+IL-1RII-, IL-1RI+IL-1RII+, and IL-1RI-IL-1RII- cells.
**Figure supplement 1.** IL-1β-mediated changes in pathogenic Th17-cell-associated gene signature of TCR-activated memory CD4⁺ T cells in humans.
**Figure supplement 2.** Effect of IL-23 on Th17 production in IL-1RI⁺IL-1RII⁻ cells and IL-1RI⁺IL-1RII⁺ memory CD4 T cells.

maturation and influences necrosis-induced sterile inflammation (*Zheng et al., 2013*). In a mouse model, IL-1RII deficiency on neutrophils and macrophages was found to aggravate K/BxN serum transfer-induced arthritis and collagen-induced arthritis by inhibiting IL-1 signals (*Martin et al., 2017*; *Shimizu et al., 2015*). However, the role of IL-RII on T cells remains to be understood.

We found that TCR-mediated induction of IL-1RI kinetically precedes IL-1RII expression by 12–24 hr and the induction of IL-1RII is predominantly observed in IL-1RI-expressing cells (*Figure 2C and D*). This suggests *cis*-regulation by IL-1RII involves its competitive binding to IL-1 and IL-1RAcP on CD4⁺ T cells under inflammatory conditions. Furthermore, treatment with IL-1β, but not other cytokines, further increased the expression of IL-1RII in TCR-stimulated CD4⁺ memory T cells, supporting the idea that there is a negative feedback loop of IL-1 signaling in CD4⁺ T cells (*Figure 2—figure supplement 2B*).

Of note, IL-1RII expression on memory CD4⁺ T cells was found to correlate with the dose of anti-CD3 mAb used for stimulation, whereas IL-1RI expression was significantly increased following weak anti-CD3 stimulation (100 ng/ml) (*Figure 2E*). *Purvis et al., 2010* demonstrated that low-strength stimulation of human CD4⁺ T cells in a Th17-polarizing cytokine milieu strongly favors Th17 responses. They proposed that low Ca²⁺ signaling in CD4⁺ T cells under low-strength stimulation is preferential for the increase in binding of NFAT to the proximal region of the IL-17 promoter in CD4⁺ T cells, thereby enhancing IL-17 production. It should be noted that this enhancement of Th17 responses is observed only under treatment with Th17-polarizing cytokine such as IL-1β, IL-23, and TGF-β, suggesting a possible role of these cytokines in modulating IL-17 production (*Purvis et al., 2010*). Our finding may give another explanation as to why low-strength TCR stimulation favors Th17 responses in human CD4⁺ T cells, in that weak induction of IL-1RII leads to elevated IL-1β signaling, which causes increased RORγt activity and IL-17 production. Thus, we suggest that high-strength stimulation induces robust expression of IL-1RII on memory CD4⁺ T cells expressing IL-1RI and suppresses excessive IL-17 production via limiting IL-1β responsiveness (*Figure 2F*).

Considering the preferential expression of IL-1RII by TCR-induced Foxp3⁺ T cells (*Figure 3*; *Tran et al., 2009*; *Mercer et al., 2010*), it is reasonable to assume that Treg-related TFs are involved in *IL-1RII* gene expression. Foxp3 forms a multiprotein complex with many protein partners related to the regulation of transcription (*Rudra et al., 2012*). Among these Foxp3 partners, NFAT plays a key role in regulation of major Treg-related molecules such as CTLA-4, IL-2, and CD25 via direct binding of the NFAT/FOXP3 complex to the promotors of these genes (*Wu et al., 2006*; *Hermann-Kleiter and Baier, 2010*). Our findings allow the addition of IL-1RII to this list of genes. In vitro assays using the CsA, a drug targeting NFAT, show selective inhibition of IL-1RII expression on activating IL-1RI⁺ CD4⁺ T cells (*Figure 4B*). Furthermore, a VIVIT peptide derived from the calcineurin-NFAT-binding motif, PxIxIT, also significantly represses IL-1RII mRNA and protein expression (*Figure 4C and D*). Repression of NFAT activation by the VIVIT peptide is attributable to inhibition of NFAT binding to calcineurin affecting NFAT-dependent gene expression without affecting calcineurin phosphatase activity (*Aramburu et al., 1998*; *Aramburu et al., 1999*). On the contrary, enhanced induction of Foxp3 following treatment with 1,25(OH)two vitamin D3 was found to primarily augment the expression of IL-1RII and resultantly, the ratio of IL-1RII⁺ to IL-1RI⁺ activating CD4⁺ T cells was significantly increased (*Figure 4H*). This implies that 1,25(OH)two vitamin D3 causes limited IL-1β responsiveness.

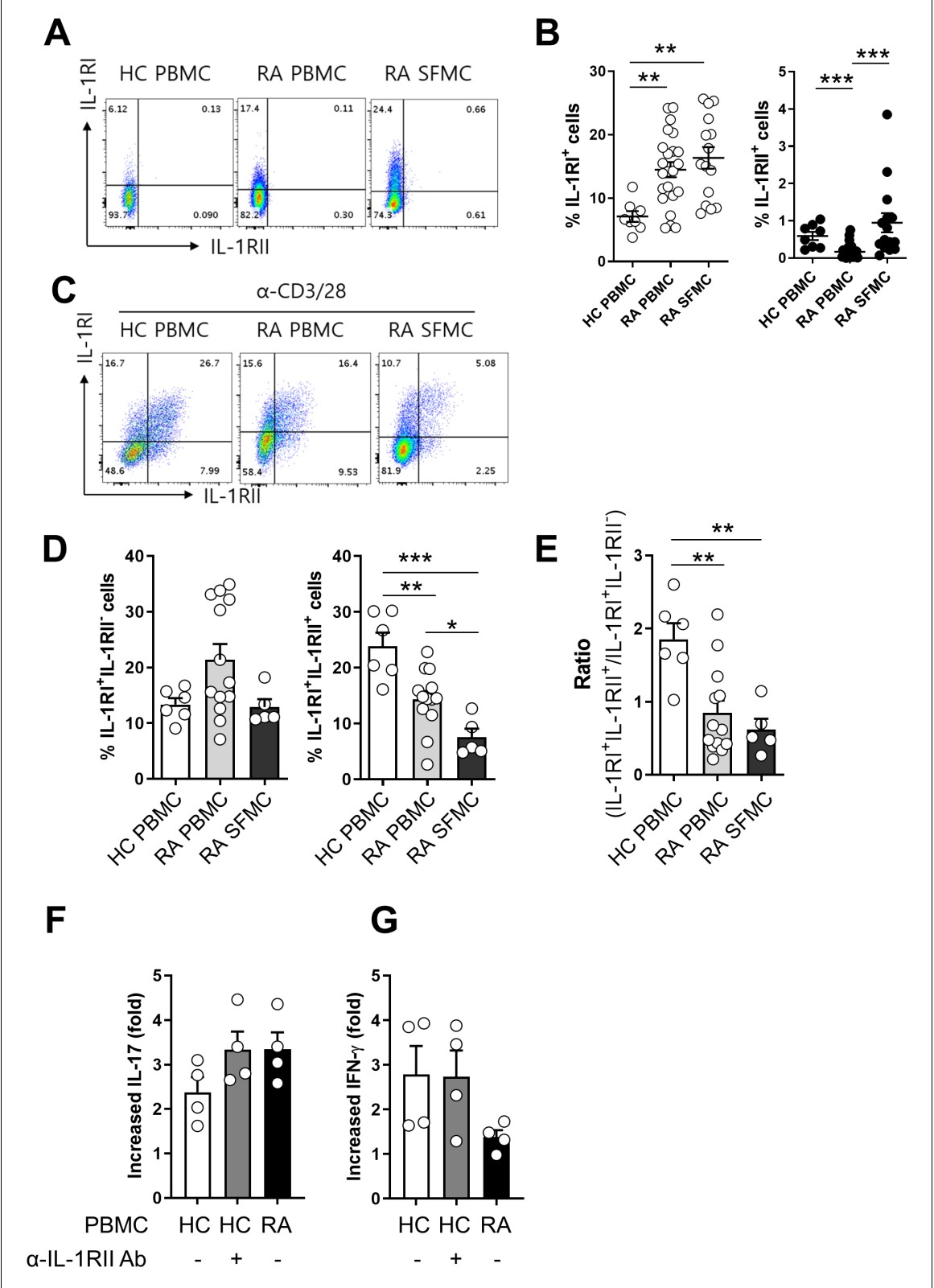

**Figure 7.** Aberrant expression of IL-1RI and IL-RII in synovial CD4 T cells in patients with rheumatoid arthritis (RA). (**A and B**) Representative flow cytometric plot (**A**) and the frequencies (**B**) of IL-1RI and IL-1RII expression on ex vivo memory CD4$^+$ T cells from peripheral blood (n = 23) and synovial fluid (n = 15) of RA patients and peripheral blood of HCs (n = 8). SFMC: synovial fluid mononuclear cells. (**C and D**) Representative flow cytometric plot (**C**) and the frequencies (**D**) of IL-1RI and IL-1RII expression on TCR-stimulated memory CD4$^+$ T cells from peripheral blood (n = 13) and synovial fluid

*Figure 7 continued on next page*

*Figure 7 continued*

(n = 5) of RA patients and peripheral blood of HCs (n = 6) at day 2 post-stimulation. (E) The ratio of IL-1RI$^+$IL-1RII$^+$ to IL-1RI$^+$IL-1RII$^-$ cells in (D). (F and G) The effect of anti-IL-1RII neutralizing Ab treatment on IL-1β-mediated IL-17 production by TCR-stimulated memory CD4$^+$ T cells of HCs (n = 4). Fold change indicates the ratio of IL-1β-mediated IL-17 production between the anti-IL-1RI Ab-treated group and control isotype-treated group. Scatter plot and bar graphs show the mean ± SEM. * = p<0.05, ** = p<0.01, and *** = p<0.001 by two-tailed uppaired or paired *t*-test.

The online version of this article includes the following source data and figure supplement(s) for figure 7:

**Source data 1.** *Figure 7B* Ex vivo expression of IL-1RI and IL-1RII on CD4+ T cells between HC PBMC, RA PBMC, and RA SFMC.

**Source data 2.** *Figure 7D* Expression of IL-1RI and IL-1RII on stimulated memory CD4+ T cells between HC PBMC, RA PBMC, and RA SFMC.

**Source data 3.** *Figure 7E* Expression of IL-1RI and IL-1RII on stimulated memory CD4+ T cells between HC PBMC, RA PBMC, and RA SFMC (ratio).

**Source data 4.** *Figure 7F and G* IL-1β-mediated IL-17 & IFN-γ production in response to TCR stimulation compared with HC and RA.

**Figure supplement 1.** Foxp3 Tregs cells in peripheral blood and synovial fluid of patients with rheumatoid arthritis (RA) and peripheral blood of HCs.

An intriguing finding in the present study is that IL-1RII expression is cooperatively controlled by NFAT and Foxp3 in memory CD4$^+$ T cells upon TCR stimulation (*Figure 5B and C*). The crystal structure analysis of the NFAT/FOXP3 complex revealed that the Wing1 domain, specifically amino acids 399–401 from FOXP3, physically interacts with the NFAT protein (*Wu et al., 2006*; *Bandukwala et al., 2011*). Based on this finding, *Lozano et al., 2015* developed an inhibitory peptide encompassing the Wing1 domain, FOXP3 393–403, which specifically interferes with the NFAT/FOXP3 interaction, without affecting the capacity of NFAT to bind DNA. This peptide inhibits Treg activity via suppression of T-cell conversion into inducible Tregs and enhancement of T cell proliferation. FOXP3 393–403, originally developed for murine Tregs, also has a similar inhibitory effect on the interaction of human NFAT/FOXP3 (*Lozano et al., 2015*). Our data show that IL-1RII induction on activating IL-1RI$^+$ CD4$^+$ T cells was significantly reduced by FOXP3 393–403 to a degree comparable to the inhibition of CD25 expression (*Figure 5A and C*).

These findings were corroborated by our luciferase reporter and ChIP assays (*Figure 5D and F*). Comparative analysis of DNA sequences using Vista tools revealed that the *IL-1RII* gene has one small CNS region (−1309 to −1203) in its promoter. Of note, potentially one Foxp3 and two consecutive NFAT binding elements are closely located at positions −1236 to −1230 and −1209 to −1188, respectively, near the CNS region (*Figure 5—figure supplement 1*). The results of the *IL-1RII* promoter assay using Foxp3-overexpressing Jurkat T cells suggest that the binding of TFs, probably the NFAT/FOXP3 complex, in the CNS region is involved in increasing *IL-1RII* promoter activity. Transfection of truncated pGL4/IL-1RII plasmids without the CNS region and of a plasmid with mutated NFAT-binding sequences in the CNS region led to a significant reduction the promoter activity (*Figure 5D*). Furthermore, it has been demonstrated that in the *IL2* promoter, the cooperative interaction between NFAT and Foxp3 involves direct protein–protein contacts between the DNA-binding domains of these transcription factors, at least some of which occur on *composite* DNA elements (ARRE sequences) with adjacent binding sites for the two transcription factors (*Wu et al., 2006*; *Bandukwala et al., 2011*; *Hu et al., 2007*). However, our ChIP data suggests that although the cooperative interaction between NFAT and Foxp3 is important for induction of IL-1RII, their major DNA-binding domains are located separately in the *IL-1RII* promoter (*Figure 5D–F*). This implies a more complex interaction occurs, possibly including tripartite partners and/or 3D looping of the genome. Furthermore, unlike the CD25 gene, there are no ARREs in the CNS region of *IL-1RII* promoter (*Wu et al., 2006*).

There are only a few studies describing IL-1RII expression on T cells. IL-1RII is preferentially expressed on TCR-stimulated Treg cells and thus, is a useful selective surface marker of activated human Treg cells (*Tran et al., 2009*; *Mercer et al., 2010*). Of interest, we found that IL-1RII expression is also induced by Treg-depleted memory CD4$^+$ T cells, suggesting that IL-1RII expression is not strictly limited to activated Tregs. Sakaguchi and colleagues have suggested that FoxP3$^+$CD4$^+$ T cells consist of three distinct subsets: CD45RA$^+$FoxP3$^{low}$ (naive Treg), CD45RA$^-$FoxP3$^{high}$ (effector Treg), and CD45RA$^-$FoxP3$^{low}$ (Foxp3-expressing non-Tregs) CD4$^+$ T cells. Of these, the CD45RA$^-$FoxP3$^{low}$CD4$^+$ T cell subset contains cells with Th17 potential (*Miyara et al., 2009*). Considering chemokine receptor and cytokine profiles of IL-1RI$^+$CD4$^+$ T cells (*Figure 3C and D*), it is possible that IL-1RI may be predominantly induced on the CD45RA$^-$FoxP3$^{low}$CD4$^+$ T-cell subset. Interestingly, IL-1RI also maintained significantly higher expression on activated human Foxp3$^+$ Tregs compared to other T cell subsets despite its important role for Th17 responses (*Tran et al., 2009*;

*Mercer et al., 2010*). Since Foxp3 is generally induced by TCR stimulation even on Treg-depleted conventional CD4+ T cells with Smad3 and NFAT dependent manner (*Figure 3—figure supplement 3*), strong TCR stimulation results in preferential expression of IL-RII, which limits IL-1β responsiveness and resultantly, reduces Th17 responses. In addition to the fact that the differentiation of Th17 and Treg cells is closely related (*Zhou et al., 2008*), both cells display instability and plasticity during immune responses; Foxp3+ Treg cells can be converted into IL-17-producing cells in certain inflammatory environments and may potentially contribute to pathogenesis of autoimmune diseases, whereas Th17 cells transdifferentiate into Treg cells during resolution of inflammation (*Gagliani et al., 2015*; *Obermajer et al., 2014*). Several studies suggest that some Th17 cells act like Tregs and this unique subset of Th17 cells are increased in tumor-bearing mice, human cancer patients (*Gagliani et al., 2015*; *Downs-Canner et al., 2017*; *Krummey et al., 2014*; *Thibaudin et al., 2016*; *Jung et al., 2017*), and during several inflammatory diseases. In our study, IL-1RI+IL-1RII+ cells were found to be similar in phenotype to Treg cells with relatively high expression of CD39, CD73, and CTLA-4 following TCR stimulation (*Figures 3B* and *6D*). Because of their phenotype and potential modulatory function via CTLA-4 and IL-1RII, IL-1RI+IL-1RII+ cells may act as pivotal regulators of the Th17-cell-mediated inflammatory environment. Thus, the function and homeostatic capacity of IL-1RI+IL-1RII+ cells require further investigation.

Rheumatoid arthritis is a prototype autoimmune disease characterized by chronic inflammation of the joint synovial membrane (*McInnes and O'Dell, 2010*; *McInnes and Schett, 2011*). The disturbed Th17/Treg balance contributes to RA pathogenesis due to the strong pro-inflammatory response of preponderant Th17 cells and impaired Treg cells lacking immunomodulatory capacity, partly due to the cytokine circumstance (*Niu et al., 2012*; *Wang et al., 2012*; *Noack and Miossec, 2014*; *Nie et al., 2013*). The interesting observation in this study was diminished TCR-mediated induction of IL-1RII, and not IL-1RI, on memory CD4+ T cells from RA patients compared to the peripheral counterpart from healthy controls (*Figure 7D*), causing a significantly reduced ratio of IL-1RI+IL-1RII+ to IL-1RI+IL-1RII- memory CD4+ T cells (*Figure 7E*). Of note, blocking of IL-1RII on memory CD4+ T cells of HCs led to increased IL-1β-mediated induction of IL-17 at a level comparable to that of RA patients (*Figure 7F*). Given the elevated level of IL-1β in RA synovial fluid and an important role of Th17 responses in RA pathogenesis (*Kim et al., 2016*; *Kay and Calabrese, 2004*), impaired induction of IL-1RII may be associated with modulation of IL-17 production in CD4+ T cells of RA patients under chronic autoreactive situations.

In summary, the current study provides new insight into the immune regulatory role of the decoy receptor, IL-1RII, expressed by human CD4+ T cells in response to TCR stimulation. Here, we investigated the molecular mechanisms underlying IL-1RII expression and analyzing its influence on Th17 responses and consequentially, the Th17/Treg balance. IL-RII expression is mediated molecularly by a cooperative complex of the NFAT and Foxp3 TFs. Further, differential expression of IL-1RI and IL-RII on activated CD4+ T cells imparts unique immunological features to these cells via modulation of IL-1β responsiveness. Together, these findings increase understanding of the modulatory mechanisms controlling the Th17 response, and of IL-1β-mediated IL-1R signaling and its contribution to the pathogenesis of autoimmune diseases such as RA.

## Materials and methods

### Cell preparation

The study protocols were approved by the institutional review board (IRB) of Seoul National University Hospital and Chungnam National University Hospital (IRB No.1109-055-378, 1306-002-491, and 1403-049-564 for Seoul National University College of Medicine/Seoul National University Hospital and IRB No.2012-01-024 for Chungnam National University Hospital). Peripheral blood of RA patients and healthy controls (HCs) was drawn after obtaining written, informed consent. The methods were performed in accordance with the approved guidelines. Peripheral blood mononuclear cells (PBMCs) were isolated from blood of RA patients and HCs by density gradient centrifugation method (Bicoll separating solution; BIOCHROM Inc, Cambridge, UK). Naive and memory CD4+ T cells were negatively purified from PBMC using human naive CD4+ T cells and human memory CD4+ T cells enrichment kit (both from STEMCELL, Vancouver, BC, Canada), respectively. In some experiments, non-Treg memory CD4+ T cells were obtained by depletion of Tregs from previously purified

memory CD4$^+$ T cells using CD4$^+$CD25$^+$CD127$^{dim/-}$ Regulatory T cells isolation kit II (Miltenyi Biotech, Bergisch Gladbach, Germany).

## Cell line

To generate Foxp3-expressing Jurkat cell line, the lentiviral vector for expression of human FoxP3 (pLEF-puro-FoxP3) was prepared as follows. pLEF-puro was created from biscistronic lentiviral vector, pLECE3, which harbors EF1alpha promoter-driven expression cassette and CMV promoter-GFP expression cassette (*Lee et al., 2019*) , by replacing GFP ORF with puroR ORF. Human FoxP3 ORF was amplified by PCR from pCMV5-entry-FoxP3 (OriGene, Rockville, MD, USA) and cloned into multi-cloning sites under the EF1alpha promoter in pLEF-puro. The lentivirus for FoxP3 expression was produced by transfecting pLEF-puro-FoxP3 plasmid into 293FT cells (Invitrogen, Waltham, MA, USA) along with packaging plasmids (pMDLg/pRRE, pRSVrev, and pMD.G). 48–72 hr after transfection, the culture supernatant containing the lentivirus was harvested and concentrated 20-fold using a centrifugal ultrafiltration unit (Amicon Ultra-15, 100kD cut-off, Merck Millipore, Burlington, MA, USA). Jurkat cells (Clone E6-1) were purchased from ATCC (Manassas, VA, USA), regularly tested to exclude mycoplasma infections, and authenticated using Short Tandem Repeat (STR) profiling.Jurkat cells were transduced with the concentrated lentivirus in the presence of polybrene (6 µg/ml, Sigma-Aldrich, St. Louis, MO, USA) via spin-infection at 2,500 rpm for 90 min at room temperature. The transduced Jurkat cells were selected in the media containing puromycin (2 µg/ml, Sigma-Aldrich, St. Louis, MO, USA) and established as a cell line by limiting dilution.

## Cell culture

Purified naive and memory CD4$^+$ T cells were cultured in serum-free X-VIVO 10 medium (Lonza, Basel, Switzerland) and RPMI 1640 medium supplemented with 10% fetal bovine serum, 1% penicillin/streptomycin, and 1% L-glutamine, respectively. Cells were seeded at $2.5 \times 10^4$ into 96-well U bottom plate and stimulated with anti-CD3/CD28-coated microbeads (Dynabeads T-Activator CD3/CD28; Thermo Fisher Scientific, Waltham, MA) for 5 or 7 days in the absence or presence of the indicated cytokines or neutralizing antibodies (Abs); recombinant human (rh) IL-1β (5 ng/ml), rhIL-6 (25 ng/ml), rhIL-23 (25 ng/ml), rhTGF-β (10 ng/ml; all from R and D systems, Minneapolis, MN), rhIL-2 (100 IU/ml, PeproTech, Rocky Hill, NJ), anti-human IL-1RII Ab, or mouse IgG2A (20 µg/ml, both from R and D systems). In some experiments, cells were stimulated with anti-CD3/CD28-coated microbeads with peptide inhibitors, chemical inhibitors, or reagents: dNP2-VIVIT and dNP2-VEET peptides were synthesized by using a solid-phase synthesis and purified by high performance liquid chromatography (Anygen, Kwangju, Korea) as previously reported (*Lee et al., 2019*). The peptides were synthesized with >95% purity. Peptide inhibitor Foxp3 393–403 (KCFVRVESEKG, 100 µM) and control peptide Foxp3 399A (KCFVRVASEKG, 100 µM) were manufactured at GenScript, Nanjing, China (*Lozano et al., 2015*; *Lim et al., 2015*), Cyclosporin A (10 or 50 nM, Sigma-Aldrich, St. Louis, MO) or 1, 25-dioxylvitamin D3 (100 µM, Sigma-Aldrich). To obtain IL-1RI$^+$IL-1RII$^-$ and IL-1RI$^+$IL-1RII$^+$ T cells, Treg-depleted memory CD4$^+$ T cells were stimulated with anti-CD3/CD28-coated microbeads for 48 hr to induce the expressions of IL-1RI and IL-1RII. The stimulated cells were harvested and stained with FITC-anti-IL-1RII, PE-anti-IL-1RI (both from R and D systems), PE-Cy7-anti-CD45RA, and APC-anti-CD4 (all from BD Bioscience, San Jose, CA), followed by cell-sorting into IL-1RI$^+$IL-1RII$^-$, IL-1RI$^+$IL-1RII$^+$, and IL-1RI$^-$IL-1RII$^-$ cells using FACSAria II (BD Bioscience).

## Flow cytometry

PBMC or cultured T cells were stained for 30 min at 4°C with following fluorochrome-conjugated Abs: APC-Cy7-anti-CD3, v500-anti-CD4, PE-Cy5-anti-CD25, APC-anti-CD25, PE-Cy7-anti-CD45RA, BV421-anti-CD161, (six from BD Bioscience), FITC-anti-GARP (Enzo Life Science, Farmingdale, NY), PE-Cy7-anti-GITR, APC-Cy7-anti-CD73, BV421-anti-CD39, PerCP-Cy5.5-anti-CTLA-4, PE-Cy7-anti-CXCR3 (five from BioLegend, San Diego, CA), PE-anti-IL-1RI, APC-anti-IL-1RII, and FITC-anti-IL-1RII (three from R and D systems). For intracellular cytokine staining (ICS), cultured T cells were re-stimulated for 6 hr with PMA (50 ng/ml) and ionomycin (1 µg/ml) in the presence of BFA for last 4 hr, followed by staining with PE-anti-IL-1RI and FITC-anti-IL-1RII Abs. The stained cells were fixed and permeabilized using Fix/Perm buffer (BioLegend), followed by staining with anti-IL-17A, anti-IFN-γ

(both from eBioscience), and anti-Foxp3 (Biolegend) Abs. The cells were acquired using a BD LSRFortessa (BD Bioscience) and analyzed using FlowJo software (Tree Star, Ashland, OR).

## Enzyme-linked immunosorbent assay (ELISA)

The amounts of IL-17, IFN-γ, and IL-10 in culture supernatant were quantified using commercially available human ELISA kits according to manufacturer's instructions (Both IL-17A ELISA and IL-10 ELISA kits from eBioscience and Human IFN-γ ELISA MAX Deluxe from BioLegend). Measurement of OD (optical density) was performed using Infinite M200 Pro Multimode microplate reader (Tecan, Männedorf, Switzerland).

## Quantitative real-time PCR

Total RNA was extracted from purified or cultured T cells with TRIzol reagent (Invitrogen) and cDNA was synthesized using GoScript Reverse Transcription System (Promega, Madison, WI). Real-time quantitative RT-PCR was performed on CFX system (Bio-Rad, Hercules, CA, USA) using SYBR green master mix (Bio-Rad). Primers were designed with Primer designing tool-NCBI (https://www.ncbi.nlm.nih.gov/tools/primer-blast/) or adopted from previously published primer sequences; Foxp3, 5'-GCACCTTCCCAAATCCCAGT-3' and 5'-GGCCACTTGCAGACACCAT-3'; IL-1RI, 5'- GTGATTGTGAGCCCAGCTA-3' and 5'-TGTTTGCAGGATTTTCCACA-3'; IL-1RII, 5'-CCTTGTCAACCTCTGGGGTA-3, and 5'-ACAGCGGTAATAGCCAGCAT-3'; IL-10, 5'- AGCTCCAAGAGAAAGGCATCT-3, and 5'- TATAGAGTCGCCACCCTGATGT-3'; IL-22, 5'- GCTTGACAAGTCCAACTTCCA-3, and 5'- GCTCACTCATACTGACTCCGTG-3'; IL-17, 5'- AACTCATCCATCCCCAGTTG-3, and 5'- GAGGACCTTTTGGGATTGGT-3'; IFN-γ, 5'- TTTGGGTTCTCTTGGCTGTT-3, and 5'- TCTTTTGGATGCTCTGGTCA-3'. The levels of gene expression were normalized to the expression of ACTINB. The comparative Ct method (ΔΔCt) was used for quantification of gene expression.

## Luciferase reporter assay

Putative NFAT and Foxp3-binding sites were screened using Promo 3.0 software (ALGGEN, http://alggen.lsi.upc.es). Conserved noncoding sequence (CNS) was analyzed using VISTA tool (http://genome.lbl.gov/vista/index.shtml). The promoter region (−1311 to +100) of human IL-1RII (Gene ID: 7850) gene containing both putative NFAT/FOXP3 binding motif and CNS (−1306 to −1200) was cloned into pGL4.10 basic luciferase reporter vector (pGL4/IL-1RII-a1414). Mutant IL-1RII reporter plasmid (pGL4/IL-1RII-a1414mu; NFAT-binding motif ACAGTTTCCA into ACAGAAACCA in the CNS region) was generated with QuikChange II Site-Directed Mutagenesis Kit (Agilent Technologies, Santa Clara, CA). Two truncated reporter vectors, pGL4/IL-1RII-a814 and pGL4/IL-1RII-a215 including −714 to +100 and −115 to +100 of the promoter, respectively, were also cloned. Plasmids were transfected into Jurkat cells using NEON electroporation system (Life Technology, Carlsbad, CA), followed manufacturer's instructions. Foxp3-expressing Jurkat cells or control Jurkat cells were transfected with the indicated IL-1RII promoter cloned luciferase vector and internal control Renilla pGL4.74 by NEON electroporation system (Life Technology). After resting for 5 hr, transfected cells were stimulated with PMA and Ionomycin for 48 hr. Cells were lysed and luminescence was detected using Dual-Luciferase Reporter Assay system (Promega) followed manufacturer's instructions.

## Chromatin immunoprecipitation (ChIP)-qPCR assay

Purified memory CD4+ T cells were stimulated with plate-bound anti-CD3 (10 μg/ml) and soluble anti-CD28 (2 μg/ml; both from Life Technology) Ab for 24 hr. Cells were cross-linked using 1% formaldehyde for 10 min and lysed with 1% SDS-lysis buffer. ChIP assay were performed using commercially available Ez-ChIP assay kit (Merck, Darmstadt, Germany) according to manufacturer's instructions. DNA-bound proteins were immunoprecipitated using anti-NFATc2 (Santa-Cruz Biotechnology, Heidelberg, Germany) or anti-Foxp3 (Thermo Fisher Scientific) Ab. Co-precipitated DNA were purified through NucleoSpin gDNA clean-up kit (MACHEREY-NAGEL, Düren, Germany) and used as a template for real-time qRT-PCR using primers as following; NFAT_B1, 5'- ATTTACCCTTTGCTCAACAAACTCC- 3' and 5'- AGTCCAGACATTTGAGGATGGGG-3'; NFAT_B2, 5'-GAGAGGTCGCAGGGGATGAT- 3' and 5'- TCTGAGAACTCATGGTCCTGG-3'; NFAT_B3, 5'- GGTGATCATGTACTCAGACCCA- 3' and 5'-GGGAGCTGGGATTTCCTAACC-3'; NFAT_BC (binding control), 5'- TGCTCTAACTATGCAATGGCT- 3' and 5'- TGATCAGAGCTCTCTTCTGATTATT-3';

Foxp3_B1, 5'- TAGGGTGGGTGGTTGAGGG - 3' and 5'- GAAGATAAAACCTCAGCGTCACC −3';
Foxp3_B2, 5'- ATTTACCCTTTGCTCAACAAACTCC- 3' and 5'- AGTCCAGACATTTGAGGATGGGG-
3'; Foxp3_B3, 5'- AGAGAGGAGCAACCAACAACA- 3' and 5'-CTTCTGAAAGTGGGAATCCAGC-3';
Foxp3_BC (binding control), 5'- AGATTCTACTAGACAAGGGCACAG- 3' and 5'-CATACAAACCAAC
TGTTCTCTCCC-3' (*Zhang et al., 2013*).

## Statistical analysis

Two-tailed paired or unpaired *t* test was done to analyze data using Prism eight software (Software, La Jolla, CA, USA) or and Microsoft Excel 2016 as indicated in the figure legends. p-Values of less than 0.05 considered statistically significant.

# Acknowledgements

The authors thank Jiyeon Jang (Seoul National University College of Medicine) for assisting in the recruitment of human subjects and thank Core Lab, Clinical Trials Center, Seoul National University Hospital for drawing blood.

# Additional information

### Funding

| Funder | Grant reference number | Author |
| --- | --- | --- |
| National Research Foundation of Korea | 2013R1A1A2012522 | Won-Woo Lee |
| National Research Foundation of Korea | 2018R1A2B2006310 | Won-Woo Lee |
| Seoul National University Hospital | 0320180220: 2018-1293 | Won-Woo Lee |

The funders had no role in study design, data collection and interpretation, or the decision to submit the work for publication.

### Author contributions

Dong Hyun Kim, Data curation, Software, Formal analysis, Investigation, Writing - original draft; Hee Young Kim, Sunjung Cho, Data curation, Formal analysis, Investigation, Methodology; Su-Jin Yoo, Resources, Formal analysis, Investigation; Won-Ju Kim, Formal analysis, Investigation, Methodology; Hye Ran Yeon, Resources, Investigation, Methodology; Kyungho Choi, Conceptualization, Resources, Formal analysis, Methodology, Writing - review and editing; Je-Min Choi, Conceptualization, Investigation, Methodology, Writing - review and editing; Seong Wook Kang, Conceptualization, Resources, Data curation, Investigation; Won-Woo Lee, Conceptualization, Resources, Data curation, Formal analysis, Supervision, Funding acquisition, Validation, Investigation, Writing - original draft, Project administration, Writing - review and editing

### Author ORCIDs

Dong Hyun Kim (ID) https://orcid.org/0000-0002-2765-4376
Won-Woo Lee (ID) https://orcid.org/0000-0002-5347-9591

### Ethics

Human subjects: The study protocols were approved by the institutional review board (IRB) of Seoul National University Hospital and Chungnam National University Hospital (IRB No.1109- 055-378, 1306-002-491, and 1403-049-564 for Seoul National University College of Medicine/Seoul National University Hospital and IRB No.2012-01-024 for Chungnam National University Hospital). Peripheral blood ofrheumatoid arthritispatients and healthy controls was drawn after obtaining written, informed consent.

Decision letter and Author response
Decision letter https://doi.org/10.7554/eLife.61841.sa1
Author response https://doi.org/10.7554/eLife.61841.sa2

## Additional files

### Supplementary files
• Transparent reporting form

### Data availability
All data generated or analysed during this study are included in the manuscript and supporting files.

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

# Appendix 1

**Appendix 1—key resources table**

| Reagent type (species) or resource | Designation | Source or reference | Identifiers | Additional information |
|---|---|---|---|---|
| Cell line (Homo-sapiens) | Jurkat (clone E6-1) | ATCC | Cat# TIB-152, RRID:CVCL_0367 | |
| Cell line (Homo-sapiens) | Foxp3+ Jurkat | This paper | | Foxp3-expressing cell line |
| Transfected construct (Homo-sapiens) | pLECE3 | *Lee et al., 2020* | | Lentiviral expression vector |
| Biological sample (Homo-sapiens) | Rheumatoid Arthritis patient (Blood and Synovial fluid sample) | Chungnam National University Hospital | Department of Internal Medicine | |
| Antibody | Anti-human-NFATc2 (mouse monoclonal) | Santacruz biotech | Cat# Sc-7296, RRID:AB_628012 | ChIP (2 μg/test) |
| Antibody | Anti-human-GITR PE-Cy7 (mouse monoclonal) | BioLegend | Cat# 371223, RRID:AB_2687170 | FACS (1:50) |
| Antibody | Anti-human-CD73 APC-Cy7 (mouse monoclonal) | BioLegend | Cat# 344021, RRID:AB_2566755 | FACS (1:25) |
| Antibody | Anti-human-CD39 BV421 (mouse monoclonal) | BioLegend | Cat# 328213, RRID:AB_10933084 | FACS (1:50) |
| Antibody | Anti-human-CTLA-4 PerCP-Cy5.5 (mouse monoclonal) | BioLegend | Cat# 369607, RRID:AB_2629673 | FACS (1:25) |
| Antibody | Anti-human-CXCR3 PE-Cy7 (mouse monoclonal) | BioLegend | Cat# 353719, RRID:AB_11218804 | FACS (1:50) |
| Antibody | Anti-human-CD127 BV421 (mouse monoclonal) | BioLegend | Cat# 351309, RRID:AB_10898326 | FACS (1:50) |
| Antibody | Anti-human-Foxp3 Alexa Fluor 647 (mouse monoclonal) | BioLegend | Cat# 320014, RRID:AB_439750 | FACS (1:25) |
| Antibody | Anti-human-IL-10 PE (mouse monoclonal) | eBioscience | Cat# 12-7108-81, RRID:AB_466178 | FACS (1:25) |
| Antibody | Anti-human-IL-17 PE-Cy7 (mouse monoclonal) | eBioscience | Cat# 25-7179-41, RRID:AB_11042972 | FACS (1:25) |
| Antibody | Anti-human-IFN-γ Alexa Fluor 700 (mouse monoclonal) | eBioscience | Cat# 56-7319-42, RRID:AB_2574509 | FACS (1:200) |
| Antibody | Anti- Human IL-23R Biotinylated (Goat polyclonal) | R and D systems | Cat# BAF1400, RRID:AB_355982 | FACS (1:50) |
| Antibody | Anti-human IL-1 RI PE (Goat polyclonal) | R and D systems | Cat# FAB269P, RRID:AB_2124912 | FACS (1:10) |
| Antibody | Anti-Human IL-1 RII Fluorescein-conjugated (mouse monoclonal) | R and D systems | Cat# FAB663F, RRID:AB_1964612 | FACS (1:10) |

*Continued on next page*

*Appendix 1—key resources table continued*

| Reagent type (species) or resource | Designation | Source or reference | Identifiers | Additional information |
|---|---|---|---|---|
| Antibody | Anti- Human IL-1RII (mouse monoclonal) | R and D systems | Cat# MAB263, RRID:AB_2125174 | Neutralization (20 µg/ml) |
| Antibody | Mouse IgG2A Isotype control (mouse monoclonal) | R and D systems | Cat# MAB003, RRID:AB_357345 | Neutralization control (20 µg/ml) |
| Antibody | Anti-human-CD4 v500 (mouse monoclonal) | BD Bioscience | Cat# 560768, RRID:AB_1937323 | FACS (1:50) |
| Antibody | Anti-human-CD4 APC (mouse monoclonal) | BD Bioscience | Cat# 555349, RRID:AB_398593 | FACS (1:25) |
| Antibody | Anti-human-CD3 APC-Cy7 (mouse monoclonal) | BD Bioscience | Cat# 557832, RRID:AB_396890 | FACS (1:50) |
| Antibody | Anti-human-CD25 PE-Cy5 (mouse monoclonal) | BD Bioscience | Cat# 555433, RRID:AB_395827 | FACS (1:25) |
| Antibody | Anti-human-CD25 APC (mouse monoclonal) | BD Bioscience | Cat# 561399, RRID:AB_10643029 | FACS (1:25) |
| Antibody | Anti-human-CD45RA PE-Cy7 (mouse monoclonal) | BD Bioscience | Cat# 560675, RRID:AB_1727498 | FACS (1:50) |
| Antibody | Anti-human-CD161 BV421 (mouse monoclonal) | BD Bioscience | Cat# 562615, RRID:AB_2737678 | FACS (1:25) |
| Antibody | Anti-human-CCR6 APC (mouse monoclonal) | BD Bioscience | Cat# 560619, RRID:AB_1727439 | FACS (1:25) |
| Antibody | Streptavidin conjugated Pacific Blue | BD Bioscience | Cat# 560797, RRID:AB_2033992 | FACS (1:25) |
| Antibody | Streptavidin conjugated Alexa Fluor 488 | Life technologies | Cat# S11223 | FACS (1:500) |
| Antibody | anti-human CD3 functional grade purified(mouse monoclonal) | eBioscience | Cat# 16-0037-81, RRID:AB_468854 | Functional T cell assay (0.1 ~ 10 µg/ml) |
| Antibody | Anti-human CD28 Functional Grade (mouse monoclonal) | eBioscience | Cat# 16-0289-85, RRID:AB_468927 | Functional T cell assay (1.5 µg/ml) |
| Recombinant DNA reagent | pGL4/IL-1RII-a1414 (luciferase plasmid) | This paper | | Containing the IL-1RII promoter region (−1314 to +100) |
| Recombinant DNA reagent | pGL4/IL-1RII-a814 (luciferase plasmid) | This paper | | Containing the IL-1RII promoter region (−714 to +100) |
| Recombinant DNA reagent | pGL4/IL-1RII-a215 (luciferase plasmid) | This paper | | Containing the IL-1RII promoter region (−115 to +100) |
| Recombinant DNA reagent | pGL4.74 (Renilla control) | Promega | Cat# E6921 | |
| Recombinant DNA reagent | pGL4.10 (luciferase plasmid) | Promega | Cat# E6651 | |
| Recombinant DNA reagent | pLEF-puro-FoxP3 | This paper | | Foxp3 overexpression plasmid |

*Continued on next page*

*Appendix 1—key resources table continued*

| Reagent type (species) or resource | Designation | Source or reference | Identifiers | Additional information |
|---|---|---|---|---|
| Peptide, recombinant protein | Recombinant human IL-1β | R and D systems | Cat# 201-LB-005 | |
| Peptide, recombinant protein | Recombinant human IL-6 | R and D systems | Cat# 206-IL-010 | |
| Peptide, recombinant protein | Recombinant human IL-23 | R and D systems | Cat# 1290-IL | |
| Peptide, recombinant protein | Recombinant human TGF-β1 | R and D systems | Cat# 240-B | |
| Peptide, recombinant protein | Recombinant human IL-2 | PeproTech | Cat# AF-200–02 | |
| Peptide, recombinant protein | Foxp3 393–403 | *Lozano et al., 2015* (PMID:26324768) | | NFAT/Foxp3 interaction inhibition peptide |
| Peptide, recombinant protein | Foxp3 399A | *Lozano et al., 2015* (PMID:26324768) | | Control peptide |
| Peptide, recombinant protein | dNP2-VIVIT | *Lee et al., 2019* (PMID:31737742) | | NFAT inhibition peptide (1 µM) |
| Peptide, recombinant protein | dNP2-VEET | *Lee et al., 2019* (PMID:31737742) | | Control peptide (1 µM) |
| Commercial assay or kit | QuikChange II XL Site-Directed Mutagenesis Kit, 10 Rxn | genomics agilent | Cat# 200523 | |
| Commercial assay or kit | EasySep Human memory CD4 + T cell enrichment kit | STEMCELL | Cat# 19157 | |
| Commercial assay or kit | EasySep Human Naïve CD4+ T Cell Isolation Kit | STEMCELL | Cat# 19555 | |
| Commercial assay or kit | CD4+CD25+CD127dim/- Regulatory T Cell Isolation Kit II, human | Miltenyi Biotec | Cat# 130-094-775 | |
| Commercial assay or kit | CD3/CD28 activation Dynabeads | Thermo Fisher | Cat# 11161D | |
| Commercial assay or kit | EZ-ChIP | Merck Millipore | Cat# 17–371 | |
| Commercial assay or kit | NucleoSpin gDNA clean-up kit | MACHEREY-NAGEL | Cat# 740230.50 | |
| Commercial assay or kit | Dual-Luciferase Reporter Assay System | Promega | Cat# E1910 | |
| Commercial assay or kit | GoScript Reverse Transcriptase | Promega | Cat# A5001 | |
| Commercial assay or kit | Power SYBR Green Master Mix | Bio-Rad | Cat# 4367659 | |
| Commercial assay or kit | Human IL-17A ELISA Ready-set-go | eBioscience | Cat# 88-7176-88, RRID:AB_2575036 | |

*Continued on next page*

*Appendix 1—key resources table continued*

| Reagent type (species) or resource | Designation | Source or reference | Identifiers | Additional information |
|---|---|---|---|---|
| Commercial assay or kit | Human IL-10 ELISA Ready-SET-Go! | eBioscience | Cat# 88-7106-86, RRID:AB_2575004 | |
| Commercial assay or kit | Human IFN-γ ELISA MAX Deluxe | Biolegend | Cat# 430104 | |
| Commercial assay or kit | Foxp3 Fix/Perm buffer set | Biolegend | Cat# 421403 | |
| Chemical compound, drug | Cyclosprorin A | Sigma | Cat# 30024 | 1 μM |
| Chemical compound, drug | 1a,25-dihydroxyvitamin D3 | Sigma | Cat# D1530 | 10 μM |
| Chemical compound, drug | Phobol 12-myristate 13-acetate (PMA) | Sigma | Cat# p8139 | 50 ng/ml |
| Chemical compound, drug | Ionomycin calcium salt | Sigma | Cat# I0634 | 1 μg/ml |
| Sequence-based reagent | All sequences are listed in Materials and methods. | | | |
| Software, algorithm | Prism | GraphPad | RRID:SCR_002798 | |
| Software, algorithm | Promo 3.0 | ALGGEN | RRID:SCR_016926 | |
| Software, algorithm | VISTA | Joint genome institute | RRID:SCR_011808 | |
| Software, algorithm | Primer designing tool | NCBI | | |
| Other | Neon Transfection System | Thermo Fisher | | |

