## [Decision Letter]

**Acceptance summary:**

IL-1β is known to mediate human Th17 responses. This study clearly shows that expression of a IL-1 decoy receptor, IL-1RII, in IL-17-producing memory CD4^+^ T cells is mediated by NFAT/Foxp3 and regulates IL-1β-dependent IL-17 responses in humans. In addition, TCR stimulation-mediated IL-1RII expression was decreased in memory CD4^+^ T cells in synovial fluid of rheumatoid arthritis patients.

**Decision letter after peer review:**

Thank you for submitting your article "Induction of the IL-1RII Decoy Receptor by NFAT/FOXP3 Blocks IL-1β-Dependent Response of Th17 Cells" for consideration by *eLife*. Your article has been reviewed by two peer reviewers, one of whom is a member of our Board of Reviewing Editors, and the evaluation has been overseen by Satyajit Rath as the Senior Editor. The following individual involved in review of your submission has agreed to reveal their identity: Amit Awasthi (Reviewer #2).

The reviewers have discussed the reviews with one another and the Reviewing Editor has drafted this decision to help you prepare a revised submission.

Summary:

In this study, the authors showed that expression of IL-1RII in IL-17-producing memory CD4^+^ T cells is mediated by NFAT/Foxp3 and regulates IL-17 responses in humans. First, the authors confirmed that IL-1β enhances IL-17 responses in human memory CD4^+^ T cells. Then, the authors showed that CD45RA-CD4^+^ memory T cells express IL-1RI. TCR stimulation of memory CD4^+^ T cells increased IL-1RI expression at the early time point (within 12h) and then increased IL-1RII expression (after 36h). In TCR-stimulated (36h) memory T cells, IL-1β response was somewhat decreased. Treg-depleted memory CD4^+^ T cells expressed IL-1RII after TCR stimulation, and these cells also expressed Treg markers including Foxp3. Then, the authors used several inhibitors and activators and showed that NFAT and Foxp3 are required for the induction of IL-1RII. Co-operation of NFAT and Foxp3 for IL-1RII gene activation was demonstrated by luciferase assay. in vitro stimulation of IL-1RI+IL-1RII+ and IL-1RI+IL-1RII- T cells with IL-1β showed that the frequency of IL-17 production decreased and Foxp3 expression increased in IL-1RII+ T cells. Finally, the authors analyzed memory CD4^+^ T cells in synovial fluid (SF) in rheumatoid arthritis (RA) patients. IL-1RI expression was higher, TCR stimulation-mediated IL-1RII expression was decreased in memory CD4^+^ T cells in SF of RA patients.

Overall, the experiments were well performed, and the results indicate that IL-1RII is a critical molecule controlling the plasticity of human TH17 cells.

Essential revisions:

1) The authors should show whether exposure of memory T cells with IL-1β increases the expression of the genes which are crucial for pathogenic functions of Th17 cells (please refer to Lee et al., 2012). It will also be interesting to see whether regulation of IL-1RI+ IL-1RII+ cells vs. IL-1RI+ IL-1RII- in context of Th17 cells pathogenic signature.

2) Since IL-23R is very critical for the survival and expansion of Th17 cells and IL-23R predominantly express on memory Th17 cells, the authors should check the correlation of IL-23R with IL-1RI+ IL-1RII+ cells vs. IL-1RI+ IL-1RII- subset of Th17 cells. The reviewer further recommends to test the out of IL-23 exposure to IL-1RI+ IL-1RII+ cells, IL-1RI+ IL-1RII- and IL-1RI- IL-1RII- subset of Th17 cells. This correlation will be very important to bring this study to a more fruitful conclusion.

3) In Figure 3A experiments, the authors depleted Treg cells, and showed that TCR stimulation induced IL-1RII expression even in Treg-depleted memory CD4^+^ T cells. But there is a possibility that residual Treg cells, but not non-Treg memory cells, expanded and expressed IL-1RII. To exclude this possibility, the authors should show that Foxp3+ cells were efficiently decreased by this depletion protocol. In another word, the authors should show "Treg-depleted memory CD4^+^ T cells" do not express Foxp3 before stimulation.

4) The authors stated that the NFAT/Foxp3 complex binds to the B1 sites of the IL-1RII gene promoter. To show the co-operation of NFAT/Foxp3 which binds to the B1 NFAT site, the authors should analyze whether NFAT binds the B1 site in Jurkat cells without Foxp3 expression.

5) The authors showed that NFAT and Foxp3 are essential for IL-1RII expression in memory CD4^+^ T cells. IL-17-producing memory CD4^+^ T cells are not supposed to express Foxp3. In this context, the authors should show what kinds of stimuli induces Foxp3 expression in memory non-Treg CD4^+^ T cells.

6) Experiments in Figure 6 seem controversial. In Figure 3D, IL-1RI+ IL-1RII+ cells did not produce IL-17, but in Figure 6A, those cells seem to produce significant amounts of IL-17 after IL-1β stimulation.

7) In Figure 7A-C experiments, the authors should analyze the percentage of Foxp3+ Treg cells in HC PBMC, RA PBMC, and RA SFMC. It is possible that Foxp3+ Treg cells mainly responded and expressed IL-1RII, particularly in HC PBMC.

---

## [Author Response]

Essential revisions:1) The authors should show whether exposure of memory T cells with IL-1β increases the expression of the genes which are crucial for pathogenic functions of Th17 cells (please refer to Lee et al., 2012). It will also be interesting to see whether regulation of IL-1RI+ IL-1RII+ cells vs. IL-1RI+ IL-1RII- in context of Th17 cells pathogenic signature.

We appreciate the reviewer for a valuable comment. It is very important point. According to the reviewer’s comment, we examined the expression profile of pathogenic Th17 cell-associated genes using memory CD4^+^ T cells stimulated with anti-CD3/CD28 microbeads for 7 days in the presence or absence of IL-1β (1, 2). As seen in Author response image 1, pathogenic Th17 cell-associated genes including IL-22, Casp1, and several chemokines were upregulated by TCR stimulation with IL-1β. It may be thought that the IL-1β-mediated upregulation of pathogenic Th17 cell-related genes is not remarkable. However, it should be noted that our results were caused by the effect of IL-1β alone and changes in the properties of already differentiated memory cells, while the previous findings differ in that they were observed during Th17 differentiation of naive CD4^+^ T cells under optimal cytokine conditions^1^,^2^. As suggested by reviewer, we also compared the expression of Th17-cell pathogenic signature between IL-1RI^+^IL-1RII^+^ and IL-1RI^+^IL-1RII^-^ cells. Our qPCR data showed that several pathogenic Th17 cell-associated genes including IL-22, CCL3, and CSF2 were also upregulated in IL-1RI^+^ IL-1RII^-^ cells than IL-1RI^+^ IL-1RII^+^ cells (Author response image 1). We added this data into Figure 1—figure supplement 1C and Figure 6—figure supplement 1 of revised manuscript (Figure 1—figure supplement 1C and Figure 6—figure supplement 1, Results).

**Author response image 1. sa2fig1:** IL-1β-mediated changes in pathogenic and nonpathogenic gene signature of TCR-activated memory CD4^+^ T cells in humans. (A) Purified memory CD4^+^ T cells were stimulated with anti-CD3/28 coated microbeads in the presence of PBS (as vehicle) or IL-1β (5 ng/ml) for 7 days. Expression of pathogenic or nonpathogenic genes in IL-1β-treated CD4 T cells was presented relative to their expression in PBS-treated CD4 T cells. (B) Treg-depleted memory CD4^+^ T cells were stimulated for 48 h to induce the expression of IL-1RI and IL-1RII. IL-1RI^+^IL-1RII^-^ and IL-1RI^+^IL-1RII^+^ cells were purified by cell sorting and stimulated through their TCR for 5 days in the presence of IL-1β. Expression of pathogenic or nonpathogenic genes in IL-1RI^+^IL-1RII^-^ cells was presented relative to their expression in IL-1RI^+^IL-1RII^+^ cells. Bar graphs show the mean ± SEM.

2) Since IL-23R is very critical for the survival and expansion of Th17 cells and IL-23R predominantly express on memory Th17 cells, the authors should check the correlation of IL-23R with IL-1RI+ IL-1RII+ cells vs. IL-1RI+ IL-1RII- subset of Th17 cells. The reviewer further recommends to test the out of IL-23 exposure to IL-1RI+ IL-1RII+ cells, IL-1RI+ IL-1RII- and IL-1RI- IL-1RII- subset of Th17 cells. This correlation will be very important to bring this study to a more fruitful conclusion.

We appreciate the reviewer’s constructive critiques and agree the reviewer’s point. IL-23 is one of critical cytokines for the differentiation, commitment, and survival of Th17 cells. In the previous study, we clearly showed that ex vivo IL-1RI^+^ memory CD4^+^ T cells had higher expression of IL-23R than IL-1RI^-^ memory CD4^+^ T cells (3). This finding was also confirmed in the present study. However, it remains unclear whether induced IL-1RI^+^ memory CD4^+^ T cells express IL-23R or not. To this end, purified memory CD4^+^ T cells were stimulated for 2 days with anti-CD3/28 coated microbeads and their expression of IL-1RI, IL-1RII, and IL-23R was analyzed. Although the frequency of IL-23R was comparable among IL-1RI^+^IL-1RII^-^, IL-1RI^+^IL-1RII^+^, and IL-1RI^-^IL-1RII^-^ subsets, the MFI (mean fluorescent intensity) of IL-23R was significantly higher in IL-1RI^+^IL-1RII^-^ subset than other subsets. According to reviewer’s recommendation, purified memory CD4^+^ T cells were stimulated for 48 h to induce the expression of IL-1RI and IL-1RII. Three different subsets, IL-1RI^+^IL-1RII^-^, IL-1RI^+^IL-1RII^+^, and IL-1RI^-^IL-1RII^-^ cells, were purified by cell sorting and cultured for 5 days with exogenous IL-1β (5 ng/ml) and/or IL-23 (25 ng/ml). No obvious effect of IL-23 on IL-17 production was observed both IL-1RI^+^IL-1RII^-^ and IL-1RI^+^IL-1RII^+^ compared with PBS control group. Of interest, the expression of IL-1RI and IL-1RII was significantly induced on day 2 after TCR stimulation but the expression of IL-23R gradually increased until day 5, showing the induction kinetics of their receptors were different. Previous our study also showed that IL-1β had a dominant effect on IL-17 production in IL-1RI^+^ memory CD4^+^ T cells than any other Th17-promoting cytokines (4). However, we cannot rule out the possibility that the IL-17 response was enhanced, if IL-23 was added on the 5th day when IL-23R expression was increased. Further studies are needed on the expression mechanism of IL-23R and the role of IL-23 in IL-IRI^+^ Th17 cells. We added this data into Figure 2—figure supplement 1, Figure 3—figure supplement 4, and Figure 6—figure supplement 2 of revised manuscript (Results).

3) In Figure 3A experiments, the authors depleted Treg cells, and showed that TCR stimulation induced IL-1RII expression even in Treg-depleted memory CD4^+^ T cells. But there is a possibility that residual Treg cells, but not non-Treg memory cells, expanded and expressed IL-1RII. To exclude this possibility, the authors should show that Foxp3+ cells were efficiently decreased by this depletion protocol. In another word, the authors should show "Treg-depleted memory CD4^+^ T cells" do not express Foxp3 before stimulation.

We appreciate the reviewer’s constructive critiques and it is very important point. In initial stage of our study, we did check how well CD25^high^CD127^(dim/-)^ Tregs were depleted after microbeads-based purification and confirmed that CD25^high^CD127^(dim/-)^ Tregs were quite efficiently depleted (Figure 3—figure supplement 2A). As pointed out by reviewer, it is possible that Foxp3^+^ T cells still remain after the depletion because CD25^high^CD127^(dim/-)^ and Foxp3^+^ Tregs are not exactly overlapped. To respond the reviewer’s comment, we analyzed the expression of Foxp3 in Treg-depleted memory CD4^+^ T cells by intracellular staining. As seen in Figure 3—figure supplement 2B, Foxp3^+^ cells were markedly removed (over 80%) by this depletion protocol. Of note, the residual Foxp3^+^ T cells after depletion belongs to non-suppressive cytokine-producing Foxp3^low^ T cells (III in Author response image 2 and Figure 3—figure supplement 2C) by the definition of the Sakaguchi group (5). In fact, foxp3^hi^ activated Treg cells (II in Author response image 2 and Figure 3—figure supplement 2C) were completely removed by our Treg-depletion system (Author response image 2 and Figure 3—figure supplement 2C). (Results).

**Author response image 2. sa2fig2:** Efficiency of sorting-based Treg depletion from CD4^+^ T cells. Six subsets of CD4^+^ T cells defined by the expression of CD45RA and CD25: (I), CD25^++^CD45RA^+^ cells → resting Treg cells; (II), CD25^+++^CD45RA^−^ cells → activated Treg cells; (III), CD25^++^CD45RA^−^ cells → non-suppressive cytokine-producing Foxp3^low^ T cells; (IV), CD25^+^CD45RA^−^ cells; (V), CD25^−^CD45RA^−^ cells; (VI), CD25^−^CD45RA^+^ cells (Miyara et al., 2009).

4) The authors stated that the NFAT/Foxp3 complex binds to the B1 sites of the IL-1RII gene promoter. To show the co-operation of NFAT/Foxp3 which binds to the B1 NFAT site, the authors should analyze whether NFAT binds the B1 site in Jurkat cells without Foxp3 expression.

We appreciate the reviewer’s constructive suggestion. To respond the reviewer’s comment, we conducted a ChIP assay in Jurkat cells with or without Foxp3 expression for confirming co-operation of NFAT/Foxp3 which binds to the B1 NFAT site. Conventional Jurkat cells, which lack Foxp3 expression, were stimulated with PMA and ionomycin for 2 hr, followed by ChIP assay using anti-human NFAT mAb. NFAT recruitment to the DNA response element B1 site were compared with that in Foxp3-expressing Jurkat cells. NFAT recruitment to the NFAT BC (binding control), a binding site in human *IL-2* promoter region, were comparable between Jurkat cells with or without Foxp3 expression, whereas NFAT recruitment to the B1 site were significantly enriched in Foxp3-expressing Jurkat cells upon stimulation with PMA and ionomycin compared with Jurkat cells without Foxp3 expression. We added this data into Figure 5—figure supplement 3 of revised manuscript (Results).

5) The authors showed that NFAT and Foxp3 are essential for IL-1RII expression in memory CD4^+^ T cells. IL-17-producing memory CD4^+^ T cells are not supposed to express Foxp3. In this context, the authors should show what kinds of stimuli induces Foxp3 expression in memory non-Treg CD4^+^ T cells.

We thank the reviewer for a valuable comment. It is very important point. Foxp3 plays a key role in CD4^+^CD25^+^ Treg cell function in mice and represents a specific marker for these cells. However, accumulating evidence shows that the expression pattern of Foxp3 differs between mice and humans, as Foxp3 is also transiently expressed in human activated non-regulatory T cells, suggesting that Foxp3, in humans, is not sufficient to induce regulatory T cell activity or to identify Treg cells (6, 7, 8, 9) . In the previous study, we demonstrated that around 20% of Treg-depleted CD4 T cells induced the expression of Foxp3 upon TCR stimulation with IL-2 and this induction was markedly increased in the presence of 1,25 dihydroxyvitamin D, an active vitamin D3 metabolite (10). To respond the reviewer’s comment, we examined what kinds of stimuli induces Foxp3 expression in Treg-depleted memory CD4^+^ T cells. Consistent with previous our study^10^, around 20% of Treg-depleted memory CD4^+^ T cells gained Foxp3 expression on day 7 upon TCR stimulation alone. The experiments using various chemical inhibitors illustrated that among major TCR signaling pathways, NFAT is critical for Foxp3 induction in Treg-depleted memory CD4^+^ T cells. Additionally, TGF-β-mediated Smad3 pathway also contribute to Foxp3 induction. Although this TGF-β could be originated from fetal bovine serum in culture media, it should be noted that TGF-β is an abundantly and ubiquitously expressed cytokine by most cell types and tissues in vivo. On the contrary, MAPK and NF-κB pathway are not related with induction of Foxp3 expression in Treg-depleted memory CD4^+^ T cells. We added this data into Figure 3—figure supplement 3 of revised manuscript (Results and Discussion).

6) Experiments in Figure 6 seem controversial. In Figure 3D, IL-1RI+ IL-1RII+ cells did not produce IL-17, but in Figure 6A, those cells seem to produce significant amounts of IL-17 after IL-1β stimulation.

We appreciate the reviewer’s constructive critique. As pointed out by the reviewer, IL-1RI^+^IL-1RII^+^ cells in Figure 3D did not produce IL-17. The cells in Figure 3D were harvested at 48 hr after stimulation with only TCR without exogenous IL-1β to induce IL-1RI and IL-1RII expression. Of note, only part of IL-1RI-expressing CD4^+^ T cells was able to acquire IL-1RII during TCR stimulation, suggesting that these IL-1RI^+^IL-1RII^+^ cells are in favorable conditions (or precommitted to differentiate into Foxp3^+^ Treg cells) for a robust cooperation of NFAT/Foxp3 and thus acquire typical features of induced Treg under conditions without IL-1β. In the Figure 6, highly purified IL-1RI^+^IL-1RII^+^ cells were further cultured for 5 days in the presence of IL-1β to examine IL-1β-mediated T-cell responses. As shown in their kinetics (Figure 2D), IL-1RII expression might be more rapidly decreased than IL-1RI on these cells. Thus, it is likely that their effect of IL-1β is reinforced, followed by increased IL-17 production. In the future, it will be necessary to study detailed mechanisms on this using the IL-RII conditional KO mouse. Nevertheless, it should be noted that IL-1RI^+^IL-1RII^+^ cells produce less amount of IL-17 and express higher level of Foxp3 on day 5 compared to IL-1RI^+^IL-1RII^-^ cells, showing their feature of IL-17 producing Treg cells.

7) In Figure 7A-C experiments, the authors should analyze the percentage of Foxp3+ Treg cells in HC PBMC, RA PBMC, and RA SFMC. It is possible that Foxp3+ Treg cells mainly responded and expressed IL-1RII, particularly in HC PBMC.

We thank the reviewer for a valuable comment. It is certainly very critical point. According to the reviewer’s comment, we newly recruited HCs and RA patients and analyzed the percentage of Foxp3^+^ Treg cells in HC PBMC, RA PBMC, and RA SFMC. In agreement with previous reports^12^,^13^, the frequency of Foxp3^+^ CD4 T cells in RA SFMC was higher than those in PBMC of HC and RA (Figure 7—figure supplement 1). A number of studies have investigated the number, phenotype, and function of Treg cells in the peripheral blood, synovial fluid, and synovial membrane of RA patients (11, 12, 13, 14) . Although conflicting results have been reported concerning Treg cell proportion in RA peripheral blood, majority of data points out an increase of Treg cells in synovial fluid of RA patients, presumably resulting in a compensatory mechanism to counteract local inflammation (15, 16). We also examined the percentage of Foxp3^+^ Treg cell upon TCR stimulation, showing that their percentage was comparable in peripheral memory CD4^+^ T cells between HCs and RA patients (Author response image 3). However, the ratio of IL-1RI^+^IL-1RII^+^ to IL-1RI^+^IL-1RII^-^ was lower in RA patients than HCs due to increased IL-1RI and decreased IL-1RII inductions as depicted in Figure 7E of our original manuscript. Thus, defect in IL-1RII expression in RA memory CD4^+^ T cells was not likely to be caused by the increased frequency of Foxp3^+^ Treg cells. Of interest, we found that plasma and synovial fluid derived from RA patients caused a defective induction of IL-1RII on memory CD4^+^ T cells of HCs. This suggests that aberrant expression of IL-1RI and IL-RII in CD4 T cells of RA patients is attributable to their proinflammatory milieu. (Results).

**Author response image 3. sa2fig3:** Foxp3 Tregs cells in peripheral blood and synovial fluid of patients with rheumatoid arthritis (RA) and peripheral blood of HCs. (A) Representative flow cytometric plot of IL-1RI, IL-1RII, and Foxp3 expression on TCR-stimulated memory CD4^+^ T cells from peripheral blood of RA patients and peripheral blood of HCs at day 2 post-stimulation. The frequency of Foxp3+, IL-1RI^+^IL-1RII^+^, IL-1RI^+^IL-1RII^-^ cells and ratio of IL-1RI^+^IL-1RII^+^ to IL-1RI^+^IL-1RII^-^ cells. (B) Representative flow cytometric plot and frequency of IL-1RI and IL-1RII expression on TCR-stimulated memory CD4^+^ T cells from HCs with plasma and synovial fluid of RA patients and plasma of HCs at day 2 post-stimulation. Bar graphs show the mean ± SEM. * = *p* < 0.05 by two-tailed paired or unpaired *t*-test. NS indicates not significant.

References:

1) Lee Y, Awasthi A et al. Induction and molecular signature of pathogenic TH17 cells. Nat Immunol. 2012 Oct;13(10):991-9.

2) Hu D et al. Transcriptional signature of human pro-inflammatory TH17 cells identifies reduced IL10 gene expression in multiple sclerosis. Nat Commun. 2017 Nov 17;8(1):1600.

3) Lee WW et al. Regulating human Th17 cells via differential expression of IL-1 receptor. Blood. 2010 Jan 21;115(3):530-40.

4) Lee WW et al. Regulating human Th17 cells via differential expression of IL-1 receptor. Blood. 2010 Jan 21;115(3):530-40.

5) Miyara M et al. Functional delineation and differentiation dynamics of human CD4+ T cells expressing the FoxP3 transcription factor. Immunity. 2009 Jun 19;30(6):899-911.

6) Wang J. et al. Transient expression of FOXP3 in human activated nonregulatory CD4+ T cells. Eur J Immunol. 2007 Jan;37(1):129-38.

7) Allan AE et al. Activation-induced FOXP3 in human T effector cells does not suppress proliferation or cytokine production. Int. Immunol., 19 (2007), pp. 345-354.

8) Miyara M et al. Functional delineation and differentiation dynamics of human CD4+ T cells expressing the FoxP3 transcription factor. Immunity, 30 (2009), pp. 899-911.

9) Ohkura N et al. Development and maintenance of regulatory T cells. Immunity. 2013 Mar 21;38(3):414-23.

10) Kang SW et al. 1,25-Dihyroxyvitamin D3 promotes FOXP3 expression via binding to vitamin D response elements in its conserved noncoding sequence region. J Immunol. 2012 Jun 1;188(11):5276-82.

11) Möttönen M et al. CD4+CD25+ T cells with the phenotypic and functional characteristics of regulatory T cells are enriched in the synovial fluid of patients with rheumatoid arthritis. Clin Exp Immunol. 2005 May;140(2):360-7.

12) Cao D et al. FOXP3 identifies regulatory CD25bright CD4+ T cells in rheumatic joints. Scand J Immunol. 2006 Jun;63(6):444-52.

13) Jiao Z et al. Accumulation of FoxP3-expressing CD4+CD25+ T cells with distinct chemokine receptors in synovial fluid of patients with active rheumatoid arthritis. Scand J Rheumatol. 2007 Nov-Dec;36(6):428-33.

14) Moradi B et al. CD4⁺CD25⁺/highCD127low/⁻ regulatory T cells are enriched in rheumatoid arthritis and osteoarthritis joints--analysis of frequency and phenotype in synovial membrane, synovial fluid and peripheral blood. Arthritis Res Ther. 2014 Apr 17;16(2):R97.

15) Alunno A et al. Altered immunoregulation in rheumatoid arthritis: the role of regulatory T cells and proinflammatory Th17 cells and therapeutic implications. Mediators Inflamm. 2015;2015:751793.

16) Cooles FA et al. Treg cells in rheumatoid arthritis: an update. Curr Rheumatol Rep. 2013 Sep;15(9):352.